# DO MLLMS REALLY UNDERSTAND SPACE? A MATHEMATICAL REASONING EVALUATION

## ABSTRACT

Multimodal large language models (MLLMs) have achieved strong performance on perception-oriented tasks, yet their ability to perform mathematical spatial reasoning, defined as the capacity to parse and manipulate two- and three-dimensional relations, remains unclear. Humans easily solve textbook-style spatial reasoning problems with over 95% accuracy, but we find that most leading MLLMs fail to reach even 60% on the same tasks. This striking gap highlights spatial reasoning as a fundamental weakness of current models. To investigate this gap, we present *MathSpatial*, a unified framework for evaluating and improving spatial reasoning in MLLMs. *MathSpatial* includes three complementary components: (i) *MathSpatial-Bench*, a benchmark of 2K problems across three categories and eleven subtypes, designed to isolate reasoning difficulty from perceptual noise; (ii) *MathSpatial-Corpus*, a training dataset of 8K additional problems with verified solutions; and (iii) *MathSpatial-SRT*, which models reasoning as structured traces composed of three atomic operations—Correlate, Constrain, and Infer. Experiments show that fine-tuning Qwen2.5-VL-7B on *MathSpatial* achieves competitive accuracy while reducing tokens by 25%. *MathSpatial* provides the first large-scale resource that disentangles perception from reasoning, enabling precise measurement of spatial reasoning skills in MLLMs. More broadly, *MathSpatial* offers a comprehensive foundation for understanding how MLLMs handle mathematical spatial reasoning. Our code and datasets will be released upon paper acceptance.

## 1 INTRODUCTION

Multimodal large language models (MLLMs) have advanced rapidly, achieving strong performance on perception-oriented tasks such as image captioning (Sarto et al., 2025; Bucciarelli et al., 2024), visual question answering (Kuang et al., 2025; Borisova et al., 2025; Pande et al., 2025), and action recognition (Ye et al., 2025). However, whether MLLMs can truly perform spatial reasoning, the ability to parse and manipulate two- and three-dimensional relationships, remains an open question. This ability is central to higher-level cognition and underpins applications such as robotic manipulation (Yuan et al., 2025), autonomous driving (Tian et al., 2025), and embodied intelligence (Feng et al., 2025), yet we find current models frequently fail on problems that humans solve with ease, as shown in Figure 1 (a).

Existing research on spatial reasoning has made progress (Marsili et al., 2025; Zha et al., 2025; Tang et al., 2025). Some benchmarks (Ma et al., 2024; Jia et al., 2025) embed spatial tasks in visually complex scenes, testing perception-heavy reasoning pipelines, while others (Wang et al., 2024; Li et al., 2025) focus on synthetic data with limited diversity. Despite these efforts, significant limitations remain in three aspects: perceptual confounds, data scarcity, and black-box reasoning, as illustrated in Figure 1 (b). First, many benchmarks embed tasks in visually complex scenes (Ma et al., 2024; Wang et al., 2024), where rendered details introduce perceptual confounds that lead to errors prior to reasoning, obscuring the clear analysis of spatial reasoning abilities. Second, existing work has not provided large-scale, broad-coverage, and high-quality training corpora for spatial reasoning, as shown in Table 1, creating a data bottleneck that limits systematic improvement of model capabilities. Finally, most existing methods treat spatial reasoning as an end-to-end black-box mapping (Chen et al., 2024a; Ray et al., 2025), without interpretability or intermediate supervision. These challenges leave current models with a clear gap compared to human ability and restrict their potential in downstream applications.

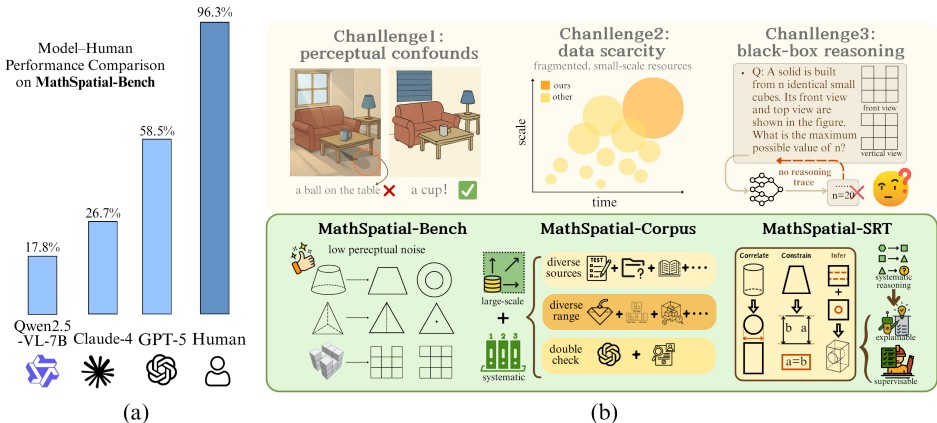

(a)                                                              (b)

Figure 1: Left: On our proposed *MathSpatial-Bench*, humans achieve over 95% accuracy while most MLLMs remain below 60%, revealing a significant capability gap. Right: The three core challenges of spatial reasoning and the design of our *MathSpatial* framework to address them.

Building on these insights, we introduce *MathSpatial*, a unified framework for spatial reasoning in multimodal large language models based on simple mathematical spatial problems. *MathSpatial* advances the field along three complementary dimensions: *rigorous evaluation*, *large-scale data*, and *interpretable methodology*, providing a systematic foundation for research and development.

To mitigate *perceptual confounds*, we design *MathSpatial-Bench*, a benchmark of 2,000 problems that isolates spatial reasoning from perception. Unlike prior datasets built on natural scenes or complex renderings (Ma et al., 2024; Wang et al., 2024), *MathSpatial-Bench* employs clean geometric problems with minimal background and texture, ensuring that errors reflect reasoning rather than visual noise. The benchmark covers three major categories—Holistic Recognition, Generative Inference, and Abstract Deduction—further divided into 11 subcategories, enabling fine-grained error analysis and systematic comparison across subcategories. *MathSpatial-Bench* reveals that while humans solve these problems with 95%+ accuracy, state-of-the-art MLLMs struggle below 60%.

To address *data scarcity*, we construct *MathSpatial-Corpus*, a large-scale dataset of 8,000 educational geometry problems. Unlike fragmented collections, it is sourced from systematic textbooks and problem banks spanning primary to high school levels. The corpus covers a wide range of tasks, from three-view recognition, unfolding and folding, and projection transformation to advanced challenges such as geometric property calculation, symmetry analysis, and causal reasoning. Each problem is accompanied by human-verified solutions, ensuring correctness and reliability, and providing task-aligned training signals to support systematic model improvement.

To overcome the limitation of *black-box reasoning*, we introduce *MathSpatial-SRT*, a framework of **S**tructured **R**easoning **T**races. It decomposes spatial problem solving in *MathSpatial* into three atomic operations: CORRELATE, which establishes correspondences across views, CONSTRAIN, which applies geometric and projection rules to enforce consistency, and INFER, which derives latent attributes or final answers. This minimal set is sufficient to cover diverse tasks while remaining interpretable and verifiable. With SRT, models learn not only to produce correct answers but also to generate reasoning traces that serve as transparent evidence for diagnosis and refinement.

The main contributions of this work are threefold:

- We introduce *MathSpatial*, the first large-scale and systematic resource for mathematical spatial reasoning in MLLMs. It consists of *MathSpatial-Bench* (2K problems) for rigorous evaluation and *MathSpatial-Corpus* (8K problems) for large-scale training, both with human-verified solutions and broad task coverage.

- We propose *MathSpatial-SRT*, a structured reasoning framework that decomposes spatial problem solving into three atomic operations—Correlate, Constrain, and Infer—providing interpretability, intermediate supervision, and transparent error diagnosis.

- Through extensive experiments, we show that a fine-tuned open-source model (Qwen2.5-VL-7B) achieves competitive performance against most closed-source systems, while reducing reasoning tokens by about 25% of the original.

Table 1: Comprehensive comparison with existing spatial/geometry reasoning benchmarks. *Bilingual*: multi-language support; *Spatial Focus*: emphasis on spatial reasoning over perception-heavy tasks; *Train Set*: availability of dedicated training data.

| Dataset | Data Domain | #Tasks | #Samples | Bilingual | Spatial Focus | Train Set |
|---|---|---|---|---|---|---|
| EmbSpatial-Bench (Du et al., 2024) | Indoor | 6 | 3,640 | ✗ | ✗ | ✓ |
| Space3D-Bench (Szymańska et al., 2024) | Indoor | 6 | 211 | ✗ | ✗ | ✗ |
| SpatialRGPT-Bench (Cheng et al., 2024) | Urban/Indoor/Sim | 12 | 1,406 | ✗ | ✗ | ✓ |
| BLINK-Spatial (Fu et al., 2024) | MSCOCO | 14 | 286 | ✗ | ✗ | ✗ |
| SpatialVLM (Chen et al., 2024a) | WebLi | 2 | 546 | ✗ | ✗ | ✓ |
| GeoEval (Zhang et al., 2024) | Educational | 3 | 5,050 | ✗ | ✗ | ✗ |
| 3DSRBench (Ma et al., 2024) | COCO/Synthetic | 4 | 6,942 | ✗ | ✗ | ✗ |
| SOLIDGEO (Wang et al., 2025a) | Educational | 8 | 3,113 | ✓ | ✓ | ✗ |
| STARE (Li et al., 2025) | Synthetic/Indoor | 3 | 3,937 | ✗ | ✗ | ✗ |
| SpatialBot-Bench (Cai et al., 2025) | COCO/VG/RTX | 5 | 200 | ✗ | ✗ | ✓ |
| VSI-Bench (Yang et al., 2025a) | Indoor | 8 | 288 | ✗ | ✗ | ✗ |
| OmniSpatial (Jia et al., 2025) | Internet | 50 | 1,387 | ✗ | ✗ | ✗ |
| **Ours** | **Educational** | **11** | **10,000** | ✓ | ✓ | ✓ |

## 2 RELATED WORK

### 2.1 MLLM SPATIAL REASONING

Multimodal large language models (MLLMs) integrate textual and visual modalities and have demonstrated remarkable potential across a wide range of vision–language tasks (Li et al., 2023; Chen et al., 2024b; Bai et al., 2025; Phan et al., 2025). Recent studies further indicate that MLLMs exhibit emerging spatial reasoning abilities, such as object-relation understanding, mental rotation, and geometric transformation (Yang et al., 2025a; Li et al., 2025; Wang et al., 2025b). These abilities are typically enhanced through supervised fine-tuning (SFT) on curated datasets. For instance, SpatialVLM (Chen et al., 2024a) and SAT (Ray et al., 2025) leverage geometric labels and spatial rules to improve mental rotation and object relation understanding, while MM-Spatial (Daxberger et al., 2025) and Spatial-MLLM (Wu et al., 2025) employ multi-view 3D scenes and large-scale corpora to strengthen 3D representation and cross-view reasoning. However, these approaches remain largely limited to free-form CoT supervision, which often produces ambiguous or inconsistent reasoning traces. In contrast, our work introduces *MathSpatial-SRT*, a structured supervision paradigm based on atomic operations (Correlate, Constrain, Infer), enabling models to learn not only accurate answers but also interpretable and verifiable reasoning processes.

### 2.2 SPATIAL REASONING BENCHMARKS

A number of benchmarks (Ma et al., 2024; Wang et al., 2024) have been proposed to evaluate spatial reasoning in MLLMs. SpatialEval (Wang et al., 2024) examines spatial relations and navigation through modality-controlled tasks; GeoEval (Zhang et al., 2024) aggregates plane, solid, and analytic geometry problems in text and text+diagram formats. Other studies focus on 3D perception: 3DSRBench (Ma et al., 2024) tests robustness across viewpoints; STARE (Li et al., 2025) uses transformation-heavy tasks like cube folding and tangrams; OmniSpatial (Jia et al., 2025) covers dynamic reasoning and perspective-taking. However, most existing studies are evaluation-only and involve high-perception scenarios, such as map layouts, 3D viewpoints, and dynamic environments (Yang et al., 2025b; Stogiannidis et al., 2025), making it difficult to assess reasoning in isolation. To address this, we propose MathSpatial, a unified ecosystem including *MathSpatial-Bench*, *MathSpatial-Corpus*, and *MathSpatial-SRT*, which reduces perceptual complexity, provides structured training resources, and supports interpretable evaluation, enabling independent assessment of spatial reasoning in MLLMs.

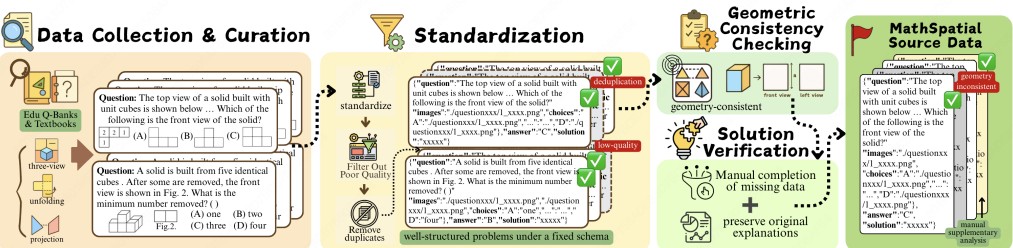

Figure 2: *MathSpatial* source data construction pipeline: Data Collection and Curation → Standardization → Geometric Consistency Checking → Solution Verification.

## 3 MATHSPATIAL: A UNIFIED FRAMEWORK FOR SPATIAL REASONING

### 3.1 OVERVIEW

In this section, we present MathSpatial, a unified framework for evaluating and advancing mathematical spatial reasoning. We first construct a dataset of 10K problems, where 2K form *MathSpatial-Bench* for comprehensive evaluation of multimodal large language models, and the remaining 8K form *MathSpatial-Corpus* for systematic training supervision. Inspired by cognitive science, we decompose spatial reasoning into three atomic operations: CORRELATE, CONSTRAIN, and INFER. Building on these operations, we introduce *MathSpatial-SRT*, which uses GPT-4o (OpenAI, 2024) to generate structured reasoning traces and applies SFT, enabling models to learn not only the final answers but also interpretable and verifiable reasoning processes.

### 3.2 MATHSPATIAL DATA CONSTRUCTION PIPELINE

To construct *MathSpatial*, we design a semi-automated pipeline that ensures both large scale and high quality. Problems are sourced from publicly available educational repositories and textbooks spanning primary to high school levels.[1] The tasks focus on key aspects of mathematical spatial reasoning, such as multi-view matching and unfolding/folding, among others.

- **Data Collection and Curation.** As illustrated in Figure 2, we systematically collect spatial reasoning problems from diverse educational sources such as Baidu Wenku, Zujuan, and other online exam banks and repositories. To mitigate dataset-level bias, we source problems across different grades, regions, and textbook editions, focusing on questions with objective numeric or multiple-choice answers. These problems are inherently formalized and structured, which reduces perceptual noise. In total, 35,428 raw candidates are systematically collected, and an initial curation step remove structurally incomplete, non-spatial, or corrupted samples, retaining 21,673 problems.
- **Preprocessing and Standardization.** Following the pipeline in Figure 2, we standardize all problems into a unified schema containing images, questions, choices, answer, and solutions. To prevent data leakage, we enforce rigorous cross-split de-duplication using MD5 hashing, GPT-4.1 vision-aided visual similarity analysis, and semantic text filtering, followed by manual verification. Low-quality items (e.g., blurry figures or ambiguous statements) are systematically discarded. This process yields approximately 11K unique, high-quality samples. Finally, Chinese source problems are translated into English via API, with quality assured through random spot-checks.
- **Geometric Consistency Checking.** As depicted in Figure 2, to guarantee geometric validity, we apply rule-based verification with human-in-the-loop review. This involves checking correspondence of length–width–height across views, enforcing dashed/solid line conventions, and validating orthographic projection rules. During this stage, approximately 0.4K geometrically inconsistent items were identified and removed. All remaining items are evaluated according to a codified geometric QA rubric, and cases for which annotators cannot reach consensus are discarded.
- **Solution Verification.** In the final stage of our pipeline (Figure 2), for problems with official solutions, we retain them directly to maintain authoritative accuracy. Among all the geometrically valid items, ∼0.8K lacked official solutions. To address this, we recruited graduate students trained

---

[1]All problems are derived from public educational materials. Copyright-sensitive items are excluded to ensure compliance.

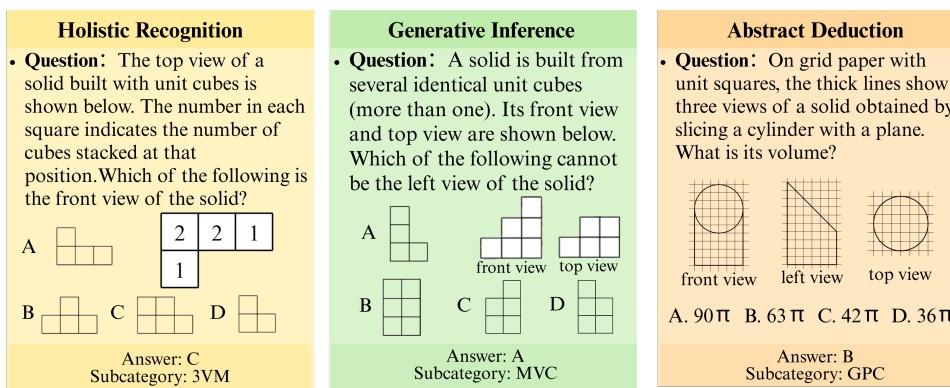

Figure 3: Examples spanning the three core categories of *MathSpatial*.

in geometry and engineering drawing to derive comprehensive solutions. Each generated solution undergoes dual-reviewer cross-validation to ensure logical completeness and reproducibility.

- **Final Output.** Following all filtering stages, 10,000 fully verified problems remain. We partition these into MathSpatial-Bench (2K) for evaluation and MathSpatial-Corpus (8K) for training. All 10K items pass a unified pipeline ensuring problem integrity, label reliability, and strict separation between the corpus and the benchmark. As shown in Table 1, MathSpatial uniquely combines bilingual support, spatial-reasoning focus without perception-heavy noise, and large-scale training supervision, making it well suited to advance multimodal spatial reasoning research.

### 3.3 MATHSPATIAL-BENCH

Building on the partition introduced above, we now focus on *MathSpatial-Bench*, the 2K-problem benchmark specifically designed to evaluate spatial reasoning in multimodal large language models. Rather than emphasizing scale, the benchmark prioritizes clarity and diagnostic value: problems are selected to minimize perceptual distractions and to isolate reasoning skills such as cross-view correspondence, geometric consistency, and deductive inference.

*MathSpatial-Bench* covers three major categories, further divided into eleven subcategories that reflect representative tasks in human education. The three categories are Holistic Recognition, Generative Inference, and Abstract Deduction. Figure 4 illustrates representative examples from each category, showing how tasks require different forms of reasoning such as multi-view correspondence, geometric consistency, and property deduction.

In total, the benchmark consists of 518 problems in Holistic Recognition, 636 in Generative Inference, and 846 in Abstract Deduction. The detailed distribution across all subcategories is shown in Figure 4, where for instance Geometric Property Calculation

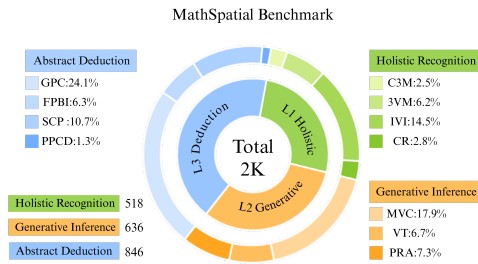

Figure 4: Distribution and composition of the *MathSpatial-Bench*

(GPC) accounts for 24.1% and Missing-View Completion (MVC) for 17.9%. This balanced coverage enables fine-grained error analysis and a comprehensive assessment of model capabilities. For detailed descriptions of each subcategory, please refer to B.2.

We only present the construction and distribution of the benchmark here. The comparative performance of humans and MLLMs on *MathSpatial-Bench* is discussed in Section 4, where we highlight the substantial gap between human accuracy (over 95%) and model performance (often below 60%).

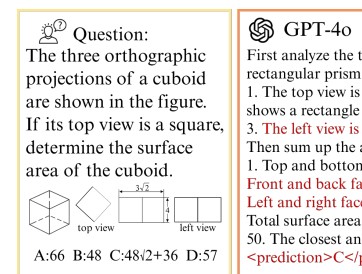

**Question:**
The three orthographic projections of a cuboid are shown in the figure. If its top view is a square, determine the surface area of the cuboid.

A:66 B:48 C:48√2+36 D:57

**GPT-4o**
First analyze the three views of the rectangular prism:
1. The top view is a square. 2. The front view shows a rectangle with height 4 and width 3√2. 3. The left view is a square with side length. Then sum up the areas of all six faces:
1. Top and bottom faces: 2 * (3 * 3) = 18; 2. Front and back faces: 2 * (3√2 * 4) = 24√23; Left and right faces: 2 * (4 * 4) = 32
Total surface area = 18 + 24√2 + 32 = 48√2 + 50. The closest answer to this is option C <prediction>C</prediction>",

**Qwen2.5-VL-7B**
First analyze the three views. The top view is a square. The front view shows a square with width 3√2 and height 4. The left view shows a rectangle.
Then get the prism's dimensions. Based on the front view, the width is 3√2 and the height is 4.The square top view gives the third dimension. Length = 3√2; Width = 3√2; Height = 4. Surface area = 2(3√2 × 3√2) + 2(3√2 × 4) + 2(3√2 × 4)= 36 + 48√2. <prediction>C</prediction>",

**MathSpatial-SRT**
\"Correlate\": From the front view, the diagonal of the square in the top view is 3√2, so its side length is 3. The front view shows the height of the prism as 4.
\"Constrain\": Each vertical face has a width of 3 (from the top view) and a height of 4 (from the front view).
\"Infer\": The four vertical faces have a total area of 4*3*4=48, and the top and bottom together add 2*3*3=18, giving a total surface area of 66 square units. <prediction>A</prediction>

Figure 5: Qualitative comparison of model outputs on a sample problem. GPT-4o and Qwen2.5-VL-7B produce free-form reasoning with inconsistencies or errors, *MathSpatial-SRT* yields a structured reasoning trace based on atomic operations (CORRELATE, CONSTRAIN, INFER), leading to a correct and interpretable solution.

## 3.4 MATHSPATIAL-CORPUS

After introducing the benchmark, we now turn to the training corpus, which provides large-scale supervision for model development. Beyond offering raw problems and solutions, *MathSpatial-Corpus* further augments each instance with structured reasoning traces, serving as richer training signals. Together with the benchmark, it completes the "training–evaluation" loop of the *MathSpatial*.

*MathSpatial-Corpus* contains more than 8,000 problems. A portion of the problems originally in Chinese is translated into English, enabling bilingual support. Each item includes a problem image, textual description, final answer, and detailed solution, ensuring completeness and reproducibility. In scale and coverage, it represents one of the largest dedicated resources for spatial reasoning.

To improve training supervision, we attach structured Chain-of-Thought (CoT) traces to every problem. These traces are constructed around three atomic operations—Correlate, Constrain, and Infer—which will be described in detail in Section 3.5. They provide explicit intermediate reasoning steps, complementing the final answer.

**Quality Assurance.** We apply a multi-stage validation pipeline to ensure the reliability of *MathSpatial-Corpus*. In addition to initial filtering, we employ GPT-4o (OpenAI, 2024) as an independent validator under a role-playing scheme, where the model alternates between the roles of reviewer and checker to identify errors, redundancies, or logical inconsistencies. Empirically, this process detects and corrects around 10% of the generated traces, substantially improving the overall quality of the structured CoT data. By doing so, *MathSpatial-Corpus* balances scale and reliability, providing a solid foundation for fine-tuning multimodal large language models on mathematical spatial reasoning. Please see Appendix B.4.2 for more details.

## 3.5 MATHSPATIAL-SRT

To move beyond black-box spatial reasoning, we design *MathSpatial-SRT*, representing spatial problem solving as sequences of atomic operations. This is inspired by decades of cognitive science showing that human spatial reasoning is not holistic but instead comprises modular steps such as feature alignment, rule application, and inference, as seen in mental rotation and spatial visualization studies (Lovett, 2012; Moen et al., 2020; Preuss & Russwinkel, 2021). We therefore decompose geometry solving into atomic operations, each modeling one cognitive facet of the process.

**Atomic operations.** We define three primitives with explicit semantics:

- CORRELATE (`corr`): establish cross-view correspondences between geometric entities (points, edges, faces).
- CONSTRAIN (`cons`): apply deterministic geometric and projection rules (e.g., visibility, alignment, hidden-line conventions).
- INFER (`infer`): deduce latent attributes or final answers from the accumulated correspondences and constraints.

We posit that this triad constitutes the *minimal sufficient set* of operations for mathematical spatial reasoning: any task in *MathSpatial* can be decomposed into a finite sequence of correlations, constraints, and inferences. Formal proofs are provided in Appendix B.4.1.

**Proposition 1** (Normal-form coverage)**.** *For every task in* MathSpatial-Bench*, there exists a finite sequence:*

$$\{\text{CORR}, \text{CONS}, \text{INFER}\}^*, \tag{1}$$

*which solves the task. Moreover, any valid reasoning process can be transformed into an equivalent sequence in this normal form.*

**Proposition 2** (Minimality)**.** *The primitive set* $\{\text{CORR}, \text{CONS}, \text{INFER}\}$ *is minimal. Removing any primitive strictly reduces expressivity:*

$$\neg\text{CORR} \;\Rightarrow\; \textit{no cross-view tasks (e.g., C3M, 3VM, MVC),} \tag{2}$$

$$\neg\text{CONS} \;\Rightarrow\; \textit{no rule admissibility checks (e.g., PRA, CR, SCP),} \tag{3}$$

$$\neg\text{INFER} \;\Rightarrow\; \textit{no numeric or decision outputs (e.g., GPC, FPBI, PPCD).} \tag{4}$$

Together, these results demonstrate that *MathSpatial-SRT* provides a sound and parsimonious foundation for spatial reasoning.

**Generation and training.** Given a problem $(x, y)$, we use GPT-4o to generate operation-level traces $r = (r_1, \ldots, r_T)$ under a constrained schema (operation type + arguments + assertion). We linearize $(r, y)$ into a single sequence and apply supervised fine-tuning. This strategy enables models to learn not only to predict correct answers but also to produce interpretable, verifiable reasoning traces that reflect the atomic operation structure. To reduce structural or logical noise, each SRT is validated through a dual-role scheme. In this process, a Reviewer agent audits every CORR/CONS/INFER step for operation-type errors, contradictions, or missing steps, and a Checker agent rewrites the trace to correct all identified issues while preserving the SRT schema. A detailed description of the validation, together with prompt templates and correction statistics, is provided in Appendix B.4.2.

**Qualitative example.** To illustrate the difference between free-form reasoning and structured traces, Figure 5 compares model outputs on the same problem. GPT-4o (OpenAI, 2024) produces a partially correct but logically inconsistent explanation, while Qwen2.5-VL-7B (Bai et al., 2025) yields fragmented reasoning with calculation mistakes. In contrast, *MathSpatial-SRT* generates a clear sequence of CORRELATE, CONSTRAIN, and INFER steps, leading to the correct answer with transparent intermediate evidence. This highlights how structured supervision not only improves accuracy but also yields interpretable reasoning traces.

## 4 EXPERIMENTS

In this section, we evaluate various models on *MathSpatial-Bench*, including our trained model *MathSpatial-SRT*. We begin by outlining the experimental setup in Section 4.1. Next, detailed evaluations on *MathSpatial-Bench* are provided in Section 4.2, comparing open-source and closed-source models across categories and difficulty levels. To validate the robustness of our approach beyond *MathSpatial*, we examine the generalization to external out-of-distribution benchmarks in Section 4.3. A systematic failure analysis follows in Section 4.4, highlighting common error patterns and challenges. Finally, ablation studies are reported in Section 4.5 to assess the contribution of different components in our framework.

### 4.1 EXPERIMENTAL SETUP

We conduct evaluations on *MathSpatial-Bench*. Our backbone models are from the Qwen2.5-VL series (Bai et al., 2025), focusing on the 3B and 7B variants to study scalability. Models are fine-tuned with supervised learning on *MathSpatial-Corpus*. Unless otherwise specified, training uses a batch size of 128, a learning rate of $1 \times 10^{-5}$, and 5 epochs. We evaluate a wide range of advanced multimodal LLMs. Closed-source systems include GPT-5 (OpenAI, 2025), GPT-4.1 OpenAI (2025), GPT-4o-2024-08-06 OpenAI (2024), Claude-3.5-Sonnet Anthropic (2024), Claude-3.7-Sonnet Anthropic (2025a), Claude-Sonnet-4 (Anthropic, 2025b), and Gemini-2.5-Pro/Flash Gemini Team (2025). Open-source baselines include Qwen2.5-VL-7B/72B Bai et al. (2025), InternVL3-8B Zhu et al. (2025), Llama3-8B Dubey et al. (2024), and GLM-4.5V Hong et al. (2025), among others. We

further evaluate our fine-tuned *MathSpatial* series models. To establish a rigorous upper bound, we recruited 80 students to evaluate the full benchmark under closed-book conditions. Each problem received 20 independent responses to ensure statistical stability, resulting in the reported micro-averaged accuracy. Detailed protocols are provided in Appendix B.3.3. For robustness, all reported results are averaged over two independent runs under identical settings. Additional details on dataset construction are included in Appendix B.2.

## 4.2 COMPARATIVE ANALYSIS ON MATHSPATIAL-BENCH

Table 2: Comprehensive evaluation on MathSpatial-Bench. Numbers denote accuracy (%), while the last column reports average token consumption. **Bold** and underline indicate the best and second-best performance, respectively. Gray background highlights our *MathSpatial* series models.

| Model | Holistic Recognition | | | | Generative Inference | | | Abstract Deduction | | | | Overall | Avg. Token |
|---|---|---|---|---|---|---|---|---|---|---|---|---|---|
| | C3M | 3VM | IVI | CR | MVC | VT | PRA | GPC | FPBI | SCP | PPCD | | |
| *Closed-Source Models* | | | | | | | | | | | | | |
| GPT-5 (OpenAI, 2025) | 28.6 | **53.7** | **66.2** | 48.2 | **61.6** | 44.8 | **71.7** | **52.3** | **60.0** | **68.1** | **57.7** | **58.5** | 676.3 |
| GPT-4.1 (OpenAI, 2025) | 2.0 | 45.5 | 43.1 | 26.8 | 24.1 | 0.0 | 55.2 | 0.0 | 29.6 | 18.3 | 46.2 | 22.6 | 676.3 |
| GPT-4o (OpenAI, 2024) | 2.0 | 36.6 | 34.1 | 33.9 | 23.5 | 1.5 | 45.5 | 0.0 | 26.4 | 16.4 | 26.9 | 19.6 | 677.4 |
| Claude 4 (Anthropic, 2025b) | 6.1 | 33.3 | 46.2 | 21.4 | 30.8 | 1.5 | 60.7 | 0.2 | 32.0 | 41.3 | 53.8 | 26.7 | 1005.5 |
| Claude 3.7 (Anthropic, 2025a) | 2.0 | 36.6 | 39.3 | 21.4 | 21.6 | 1.5 | 57.9 | 0.0 | 23.2 | 26.3 | 34.6 | 21.5 | 885.8 |
| Claude 3.5 (Anthropic, 2024) | 2.0 | 32.5 | 42.8 | 23.2 | 26.9 | 0.7 | 57.2 | 0.0 | 24.8 | 21.1 | 46.2 | 22.3 | 858.6 |
| Gemini-2.5-pro (Gemini Team, 2025) | 31.4 | 42.5 | 43.5 | 26.7 | 58.5 | 44.9 | 28.1 | 45.6 | 41.7 | 49.6 | 33.3 | 44.9 | 913.8 |
| Gemini-2.5-flash (Gemini Team, 2025) | **34.3** | 42.5 | 46.7 | 26.7 | 58.5 | **47.8** | 43.8 | 50.0 | 25.0 | 57.4 | 33.3 | 48.5 | 1115.2 |
| *Open-Source Models* | | | | | | | | | | | | | |
| Qwen2.5-VL-7B (Bai et al., 2025) | 0.0 | 32.5 | 34.8 | 17.9 | 24.4 | 4.5 | 41.4 | 0.0 | 20.0 | 13.6 | 15.4 | 17.8 | 465.3 |
| DeepSeek-VL2-7B (Wu et al., 2024) | 0.0 | 14.6 | 11.7 | 7.1 | 5.9 | 4.5 | 2.1 | 2.1 | 11.2 | 4.2 | 7.7 | 6.6 | 397.4 |
| InternVL3-8B (Zhu et al., 2025) | 0.0 | 29.3 | 35.9 | 23.2 | 21.3 | 0.7 | 40.0 | 0.0 | 19.2 | 14.6 | 15.4 | 17.4 | 473.5 |
| Llama3-8B (Dubey et al., 2024) | 10.2 | 15.4 | 20.7 | 17.9 | 15.1 | 20.9 | 23.4 | 9.3 | 13.6 | 12.2 | 7.7 | 15.0 | 785.4 |
| Idefics3-8B (Laurençon et al., 2024) | 0.0 | 22.8 | 25.9 | 14.3 | 18.2 | 1.5 | 22.1 | 0.0 | 8.0 | 6.1 | 0.0 | 11.7 | **294.5** |
| Pixtral-12B (Agrawal et al., 2024) | 4.1 | 22.8 | 34.1 | 26.8 | 20.4 | 1.5 | 34.5 | 0.0 | 8.0 | 12.2 | 3.8 | 15.3 | 692.3 |
| Kimi-VL-A3B-Thinking (Team et al., 2025) | 2.0 | 33.3 | 31.7 | **50.0** | 23.8 | 0.0 | 47.6 | 0.6 | 23.2 | 5.6 | 11.5 | 18.2 | 785.2 |
| Llama-4-Maverick-17B-128E (Singh, 2025) | 24.5 | 32.5 | 41.0 | 25.0 | 29.1 | 4.5 | 47.6 | 4.6 | 15.2 | 29.6 | 30.8 | 23.8 | 845.2 |
| Qwen2.5-VL-72B-Instruct (Bai et al., 2025) | 2.0 | 33.3 | 35.9 | 23.2 | 26.9 | 0.0 | 47.6 | 0.0 | 20.8 | 17.8 | 19.2 | 19.7 | 497.6 |
| GLM-4.5V (Hong et al., 2025) | 4.1 | 32.5 | 43.4 | 30.4 | 23.0 | 1.5 | 57.9 | 0.0 | 22.4 | 14.1 | 30.8 | 21.0 | 1391.1 |
| **MathSpatial-Qwen2.5-VL-7B** | 4.1 | 40.7 | 46.2 | 30.4 | 27.7 | 0.7 | 46.2 | 0.0 | 21.6 | 17.4 | 26.9 | 22.1 | 351.9 |
| **MathSpatial-InternVL3-8B** | 2.0 | 50.4 | 36.2 | 33.9 | 22.7 | 11.9 | 56.6 | 3.9 | 23.2 | 15.5 | 15.4 | 22.6 | 318.3 |
| **MathSpatial-Llama3-8B** | 16.3 | 30.1 | 24.5 | 41.1 | 20.4 | 30.6 | 27.6 | 10.2 | 20.8 | 16.4 | 7.7 | 20.3 | 397.3 |
| *Human Performance* | | | | | | | | | | | | | |
| Human | 100.0 | 89.4 | 97.6 | 98.2 | 96.1 | 97.0 | 95.2 | 96.5 | 94.4 | 97.7 | 96.2 | 96.3 | – |

The results in Table 2 reveal large disparities between humans, closed-source models, and open-source models on *MathSpatial-Bench*. Human accuracy exceeds 95% across all categories, while most MLLMs remain below 60%, underscoring the significant gap in spatial reasoning. Within models, closed-source systems generally outperform open-source ones, but both remain far from human-level ability, as detailed below.

- Closed-source models lead but remain far from human. Gemini-2.5-flash (48.5%) and Gemini-2.5-pro (44.9%) achieve the best accuracy, ahead of the GPT-4 series (22–23%) and Claude models (21–27%). Even GPT-5 (58.5%) reaches only about half of human performance (96.3%).
- Open-source models lag, but *MathSpatial* fine-tuning bridges the gap. Qwen2.5-VL-7B (17.8%), InternVL3-8B (17.4%), and Llama3-8B (15.0%) perform poorly, with larger variants bringing only modest gains (Qwen2.5-VL-72B 19.7%, GLM-4.5V 21.0%, Llama-4-Maverick-17B 23.8%). Our *MathSpatial-InternVL3-8B* improves to 22.6%, surpassing size-comparable open-source baselines while being the most token-efficient in this range.
- Higher-level reasoning is the main bottleneck. While Holistic Recognition reaches 40%+ for some models (e.g., Gemini-2.5-flash, *MathSpatial-Qwen2.5-VL-7B*), Generative Inference and Abstract Deduction remain difficult: tasks such as GPC and PPCD stay near chance ($\leq 33\%$).
- Clear trade-offs between accuracy and cost. While GPT-5 sets the SOTA in accuracy, high-performing models often incur heavy computational costs—most notably Gemini-2.5-flash (1115.2 tokens). In contrast, our *MathSpatial* series demonstrates superior efficiency: MathSpatial-InternVL3-8B achieves competitive open-source performance using only 318.3 tokens, less than

one-third of the consumption of leading closed-source models. Crucially, our fine-tuning reduces token consumption by $\sim$25%[2] compared to base models, fostering more concise reasoning.

## 4.3 GENERALIZATION TO EXTERNAL BENCHMARKS

To verify that *MathSpatial* series models have acquired robust spatial reasoning skills rather than merely overfitting to the training distribution, we extend our evaluation to three external out-of-distribution (OOD) benchmarks: SOLIDGEO (Wang et al., 2025a), GeoEval (Zhang et al., 2024), and the real-world subset of 3DSRBench (Ma et al., 2024).

Table 3: Performance comparison on external OOD benchmarks.

| Benchmark | Qwen2.5-VL-7B | | MathSpatial-Qwen2.5-VL-7B (Ours) | | Improvement | |
|---|---|---|---|---|---|---|
| | Acc (%)↑ | Avg. Token↓ | Acc (%)↑ | Avg. Token↓ | Acc↑ | Avg. Token↓ |
| SOLIDGEO (Wang et al., 2025a) | 15.5 | 490.2 | 18.9 | 385.6 | +3.4 | -21.3% |
| GeoEval (Zhang et al., 2024) | 17.6 | 395.4 | 19.4 | 367.5 | +1.8 | -7.1% |
| 3DSRBench (Real) (Ma et al., 2024) | 48.4 | 752.9 | 49.7 | 643.2 | +1.3 | -14.6% |

As shown in Table 3, while *MathSpatial-Bench* is designed to minimize perceptual dependencies, our results confirm that the reasoning skills acquired from *MathSpatial-Corpus* transfer effectively even to visually complex, perception-heavy domains. *MathSpatial-Qwen2.5-VL-7B* yields consistent gains (+1.3% to +3.4% accuracy) alongside a notable 7.1%–21.3% reduction in token consumption. This indicates the SRT fosters more concise reasoning patterns that persist even when the model encounters novel problem types, rather than simply memorizing dataset-specific formatting templates.

## 4.4 ERROR ANALYSIS

To better understand the limitations of current MLLMs on mathematical spatial reasoning, we conduct a fine-grained error analysis on *MathSpatial-Bench*. Errors are grouped into six categories: **(1)** Projection Error, where models misinterpret top, side, or front views; **(2)** Feature Error, such as omitting small components or inventing edges; **(3)** Scale Error,

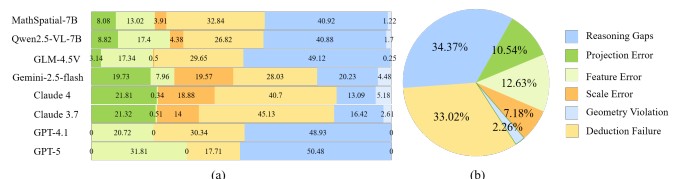

Figure 6: Fine-grained error analysis on *MathSpatial-Bench*. (a) Error frequency distribution for baselines across 6 subcategories. (b) Overall error rate breakdown by failure mode.

failing to preserve relative sizes; **(4)** Geometry Violation, where outputs break orthographic or visibility rules; **(5)** Reasoning Gaps, reflecting incomplete or inconsistent chains of thought; and **(6)** Deduction Failure, where models cannot synthesize multiple cues into a final conclusion.

Figure 6(a) shows distinct weaknesses across model families: GPT-5/4 series are dominated by reasoning gaps, Claude models often violate geometric rules, Gemini-2.5-flash suffers more from projection and scale errors, while open-source models like Qwen2.5-VL-7B and *MathSpatial-7B* [3] mainly show reasoning gaps but fewer feature errors. As summarized in Figure 6(b), the majority of errors come from **reasoning gaps (34.4%)** and **geometry violations (33.0%)**, followed by projection (12.6%) and

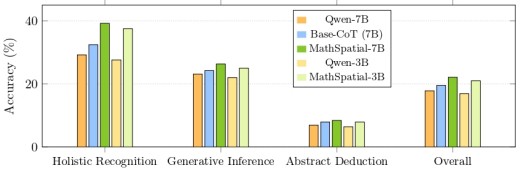

Figure 7: Ablation studies on MathSpatial across different training strategies (vanilla baseline, Base-CoT, SRT CoT) and model scales (3B, 7B).

feature errors (10.5%). Scale (7.2%) and deduction failures (2.3%) are less frequent but still harmful.

These findings highlight two core challenges: *(i)* enforcing low-level geometric consistency across views, and *(ii)* sustaining coherent multi-step reasoning chains. Both remain unsolved by current MLLMs, underscoring the necessity of structured supervision such as *MathSpatial-SRT*.

---

[2]For example, average token usage dropped from 465.3 to 351.9 for Qwen2.5-VL-7B.

[3]In Sections 4, we refer to *MathSpatial-Qwen2.5-VL-7B* as *MathSpatial-7B* for brevity.

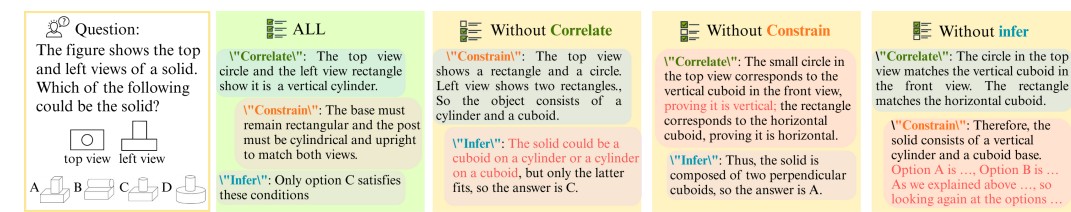

Figure 8: *MathSpatial-SRT* example: complete solution with all operations versus failures when removing Correlate, Constrain, or Infer.

## 4.5 ABLATION STUDIES

### 4.5.1 ABLATION ON TRAINING STRATEGY

As shown in Figure 7, we compare three variants: vanilla baseline (Qwen2.5-VL-7B), free-form CoT baseline (Base-CoT) [4], and our structured reasoning traces (SRT CoT). SRT CoT improves accuracy from 17.8% to 22.1%, gaining +4.3% over baseline and +2.6% over Base-CoT. The largest improvement occurs in Holistic Recognition (+10.0%), with Generative Inference (+3.2%) and Abstract Deduction (+1.5%) also benefiting. Subtask analysis (Table 2) shows 3VM, IVI, and CR improve most, indicating CORRELATE/CONSTRAIN operations effectively reduce cross-view confusion and enforce geometric consistency.

Figure 8 presents a qualitative example. With all operations enabled, the model produces correct, interpretable reasoning. Removing any operation causes distinct failures: without CORRELATE, view alignment fails; without CONSTRAIN, geometric rules are misapplied; without INFER, final synthesis fails. This shows that each atomic operation is indispensable for robust spatial reasoning.

### 4.5.2 ABLATION ON MODEL SCALE

Figure 7 (b) examines performance across model sizes. *MathSpatial-3B* outperforms its baseline ($16.9\% \rightarrow 21.0\%$, +4.1%), mirroring the 7B trend: Holistic Recognition shows largest improvement (+9.9%), with Generative Inference (+3.0%) and Abstract Deduction (+1.5%) also improving. Notable subtask gains appear in 3VM, IVI, CR, PRA, and PPCD, confirming structured supervision's robustness across scales.

## 5 CONCLUSION

In this paper, we introduced *MathSpatial*, a unified framework for studying spatial reasoning in multimodal LLMs. *MathSpatial* provides the first large-scale benchmark and corpus dedicated to mathematical spatial reasoning, together with *MathSpatial-SRT*, a structured reasoning framework based on three atomic operations. Experiments on *MathSpatial-Bench* reveal a substantial gap between human and model performance, with humans exceeding 95% accuracy while most MLLMs remain below 60%. We further show that structured supervision through *MathSpatial-SRT* enables a fine-tuned open-source model to achieve competitive accuracy against closed-source systems while reducing token usage, demonstrating both improved interpretability and efficiency.

## ETHICS STATEMENT

This work adheres to the ICLR Code of Ethics. No human subjects or animal experimentation are involved. All problems used in MathSpatial are derived from publicly available educational materials such as textbooks and online problem banks (e.g., Baidu Wenku, Zujuan). Copyright-sensitive items are excluded to ensure compliance. The dataset contains only synthetic geometry problems with images and textual statements, without any personal or sensitive information. We take care to avoid potential misuse and emphasize that MathSpatial is intended solely for academic research on mathematical spatial reasoning.

---

[4]Same generation and training pipeline to MathSpatial-SRT, but uses free-form reasoning traces.

## REPRODUCIBILITY STATEMENT

We make every effort to ensure the reproducibility of our results. The MathSpatial-Bench (2K) and MathSpatial-Corpus (8K) datasets, along with all code for preprocessing, training, and evaluation, will be released in an anonymous repository. Detailed experimental settings, including model architectures, training hyperparameters, and hardware configurations, are fully documented in the paper and supplementary material. We also provide structured reasoning traces (MathSpatial-SRT) to facilitate supervised fine-tuning and evaluation. Since all problems are sourced from public educational repositories, they can be independently verified, ensuring consistency of benchmarks and training data. We believe these measures will enable other researchers to reproduce our results and build upon MathSpatial to advance spatial reasoning research in multimodal LLMs.

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

## A  LARGE MODEL USAGE

In line with the ICLR 2026 policies on large language model (LLM) usage, we explicitly disclose how LLMs were used in the preparation of this submission.

**Use for Paper Writing.** During the writing stage, we employed large language models (including GPT-5, GPT-4.1 via API) to assist with improving grammar, refining wording, and generating preliminary drafts of certain paragraphs. All such content was subsequently reviewed, verified, and substantially revised by the authors to ensure accuracy, originality, and alignment with the scientific contributions of this work. No section of the paper was left unedited or unverified by the authors.

**Author Responsibility.** Consistent with Policy 2, the authors take full responsibility for all claims, analyses, and conclusions in the paper. The models were used only as aids to enhance clarity and efficiency in writing, not as independent sources of factual content. All experimental design, data collection, analysis, and final interpretations were performed by the authors.

**Scope of Usage.** The use of LLMs in this work was restricted to paper writing assistance. They were not used to generate results, derive proofs, or conduct analysis without human verification. The disclosure here, as well as in the submission form, fulfills the ICLR requirement that all contributions of LLMs be acknowledged transparently.

## B  APPENDIX

### B.1  ANNOTATION GUIDANCE

To complement the main pipeline description (Figure 2), we provide detailed guidance on how annotation was conducted in practice. This section elaborates on the interfaces, annotator workflow, and validation mechanisms to ensure that the resulting dataset is both reliable and reproducible.

#### B.1.1  DATA COLLECTION AND CURATION

**Source Diversity and Bias Mitigation.** Problems were systematically sourced from multiple online repositories, including Baidu Wenku, Zujuan, and provincial exam archives. This strategy ensures broad coverage across difficulty levels and problem styles, effectively mitigating stylistic and regional bias. To avoid subjective labeling bias, we strictly include only problems with objective numeric or multiple-choice answers.

**Initial Filtering.** A total of *35,428* raw candidates were collected. Annotators systematically removed structurally incomplete pages, non-spatial questions (e.g., arithmetic, logic puzzles), and corrupted scans or duplicated PDF segments. After this rigorous filtering, *21,673* spatially valid items remained, corresponding to a *61.1%* retention rate.

**Metadata Recording.** Each retained problem was tagged with rich metadata, including its origin, difficulty level (easy/medium/hard), and specific geometric topic (e.g., orthographic projection, 3D geometry, view-matching).

### B.1.2 PREPROCESSING AND STANDARDIZATION

**Unified Schema.** Every item was standardized into a five-field JSON structure: {images, questions, choices, answer, solution}.

**Duplicate Detection.** We applied a rigorous multi-stage filtering pipeline combining hashing and semantic analysis. For image-level detection, we employed MD5 hashing for exact duplicates and GPT-4.1 vision similarity scoring for diagrams differing in resolution or scaling. For text-level detection, we utilized Sentence-BERT encodings. Let $T_i$ and $T_j$ be the textual content of two problems, and $\text{sim}(T_i, T_j)$ be their cosine similarity. The filtering action $A$ is defined as:

$$A(T_i, T_j) = \begin{cases} \text{Discard (High-conf.)} & \text{if } \text{sim}(T_i, T_j) \geq 0.90 \\ \text{Manual Review} & \text{if } 0.85 \leq \text{sim}(T_i, T_j) < 0.90 \\ \text{Retain} & \text{if } \text{sim}(T_i, T_j) < 0.85 \end{cases} \tag{5}$$

Following this automated screening, annotators reviewed approximately *1K* cross-set candidate pairs, removing semantically equivalent statements or paraphrased variants. After preprocessing, approximately *11K* unique and high-quality items remained.

**Language Standardization.** Original Chinese problems were translated into English using a commercial API. To ensure terminology consistency, annotators performed spot-checks on approximately *20%* of the translated items.

### B.1.3 GEOMETRIC CONSISTENCY CHECKING

**Rule-Based Verification.** Automated scripts enforced three core geometric constraints: dimensional correspondence across views (Front-Top-Side length/width/height alignment), correct usage of dashed/solid lines for hidden/visible edges, and orthographic projection compliance.

**Human-in-the-Loop Review.** Approximately *1K* items were flagged by the validator as potentially inconsistent. Annotators inspected all flagged cases; roughly *60%* were correctable and repaired, while the remaining ∼*0.4K* showed irreconcilable inconsistencies and were discarded after senior review.

**Codified QA Rubric & Consensus.** All surviving items were evaluated according to a standardized geometric QA rubric governing view correspondence logic and admissible projection conventions. Items where annotators could not reach consensus were strictly discarded.

### B.1.4 SOLUTION VERIFICATION

**Cases With Official Solutions.** For problems with authoritative solutions, we retained them directly after formatting verification to maintain accuracy.

**Cases Without Solutions.** For the subset of items lacking official solutions, we implemented a strict generation protocol. Six graduate annotators with backgrounds in geometry or engineering drawing independently solved assigned subsets. Each solution was subsequently cross-checked by two additional annotators, and only problems with unanimous agreement were accepted. The protocol required full justification of intermediate reasoning steps and the use of explicit spatial correspondences.

Table 4: Organization of the *MathSpatial Benchmark* (2000 problems). Each subcategory lists the number of problems (#) and proportion (%). The benchmark spans three levels: Holistic Recognition, Generative Inference, and Abstract Deduction.

| Level | Subcategory | Brief Description | # | (%) |
|---|---|---|---|---|
| Holistic Recognition | C3M | Complex 3-view → 3D matching; select the unique 3D model consistent with three views (hard distractors: symmetry, hidden/occluded features). | 49 | 2.5 |
| | 3VM | 3D model → 3-view matching; given a 3D model, choose the valid set of orthographic projections. | 123 | 6.2 |
| | IVI | Incorrect view identification; detect the wrong (or only correct) view given a 3D object or a pair of correct views. | 290 | 14.5 |
| | CR | Component recognition; identify the correct Boolean composition of complex structures. | 56 | 2.8 |
| Generative Inference | MVC | Missing-view completion; infer a missing orthographic view from given ones under standard projection rules. | 357 | 17.9 |
| | VT | Viewpoint transformation; predict the target-perspective view (tests mental rotation and depth consistency). | 134 | 6.7 |
| | PRA | Projection-rule application; enforce dashed/solid line conventions and visibility across views. | 145 | 7.3 |
| Abstract Deduction | GPC | Geometric property calculation; compute volume, area, angle, centroid, etc., from multi-view or 3D cues. | 482 | 24.1 |
| | FPBI | Function & physical behavior inference; predict stability, support, rolling tendency, or interlock. | 125 | 6.3 |
| | SCP | Structural change prediction; reason about effects of edits (add/remove parts, cutouts, holes) on views or properties. | 213 | 10.7 |
| | PPCD | Path planning & collision detection; infer feasible paths and collision outcomes in constrained layouts. | 26 | 1.3 |

### B.1.5 FINAL DATASET OUTPUT

**Corpus Scale and Partitioning.** The final dataset consists of over *10K* fully curated problems. We release two subsets: *MathSpatial-Bench* (*2K* items) for systematic evaluation and *MathSpatial-Corpus* (*8K* items) for supervised training.

**Unique Properties.** MathSpatial stands out as the only dataset to combine a focused scope on spatial reasoning (minimizing perception-heavy clutter) with authoritative solution annotations at scale.

## B.2 DATASET

To provide a detailed understanding of **MathSpatial-Bench**, we first present its overall composition and subcategory definitions, followed by representative examples from each subcategory. Together, these provide both a systematic and an intuitive view of the benchmark.

### B.2.1 SUBCATEGORY DESCRIPTIONS

We organized the 2000 problems into 11 fine-grained subcategories, grouped under three reasoning levels: *Holistic Recognition*, *Generative Inference*, and *Abstract Deduction*. Table 4 summarizes each subcategory with its name, brief description, problem count, and proportion, providing a structured overview of the dataset's coverage and distribution.

### B.2.2 CASE STUDIES

To further illustrate the dataset, we provide one representative problem from each subcategory. As shown in Figure 9, the examples are arranged in three columns, corresponding to the three reasoning levels (*Holistic Recognition*, *Generative Inference*, *Abstract Deduction*), allowing readers to visually grasp the diversity of problem types and the nature of the reasoning challenges in each category.

### B.3 More Details about Results

#### B.3.1 Comparison Results Analysis

In this section, we provide a detailed performance analysis of state-of-the-art models on MathSpatial-Bench, covering a diverse range of both closed-source and open-source systems. **(1)** Among closed-source models, GPT-5 achieves the best overall performance (58.5%), yet still falls far short of human-level accuracy (96.3%). Gemini-2.5-flash and Gemini-2.5-pro follow (48.5% and 44.9%), consistently outperforming the GPT-4 and Claude series (19–27%). **(2)** Open-source models perform worse overall, with Qwen2.5-VL-7B, InternVL3-8B, and Llama3-8B scoring between 15–18%. Scaling to Qwen2.5-VL-72B and GLM-4.5V yields modest improvements (19.7% and 20.9%). **(3)** Our fine-tuned MathSpatial-Qwen2.5-VL-7B achieves 22.1%, outperforming all other open-source baselines and approaching weaker closed-source systems, while being more efficient in token usage. These results highlight both the difficulty of MathSpatial-Bench and the value of structured supervision in closing the gap between open-source and proprietary MLLMs.

#### B.3.2 Ablation Studies Analysis

To better understand the contribution of different design choices, we compare five representative variants: Qwen2.5-VL-7B, Base-CoT, *MathSpatial-7B*, Qwen2.5-VL-3B, and *MathSpatial-3B*. Figure 7 reports performance across the three major task groups—Holistic Recognition, Generative Inference, and Abstract Deduction.

Across Holistic Recognition, we observe that MathSpatial-tuned models consistently yield higher accuracy on fine-grained categories such as IVI and CR, outperforming both the Base-CoT baseline and the smaller Qwen2.5-VL-3B variant. This indicates that explicit structural supervision substantially enhances models' ability to align multi-view and holistic object representations. In Generative Inference, differences among models are less pronounced, with PRA dominating performance variation. Still, *MathSpatial-7B* provides a modest but stable improvement, suggesting that enhanced spatial priors benefit even when generation requires synthesis beyond perception. For Abstract Deduction, the advantage of targeted spatial reasoning training becomes more evident. Both *MathSpatial-7B* and *MathSpatial-3B* achieve stronger gains on FPBI and SCP, categories that demand multi-step logical consistency, while Base-CoT and Qwen2.5-VL baselines remain limited. This highlights the importance of SRT and domain-specific supervision in supporting deductive reasoning.

Overall, the ablation results show that spatially-focused training leads to systematic improvements across all three levels, with larger gains in tasks that require compositional deduction rather than local recognition alone.

#### B.3.3 Human Evaluation Protocol

We established the human baseline through a systematic evaluation protocol rather than ad-hoc sampling to ensure the statistical significance and reproducibility of the 96.3% score. The exact specifications of our study are as follows.

**Participants and Demographics.** We recruited $N_p = 80$ middle and high school students from standard academic tracks who had completed the corresponding mathematics curriculum but received no task-specific training. This setup ensures that the human baseline reflects typical student-level spatial reasoning ability, rather than expert-level annotation or test familiarity.

**Scale and Redundancy.** To ensure statistical stability, we enforced a high redundancy coverage factor ($\mathcal{C}$). Given the total number of problems in the benchmark $N_{\text{total}} = 2,000$, and each participant solving a subset $N_{\text{sub}} = 500$, the coverage is calculated as:

$$\mathcal{C} = \frac{N_p \times N_{\text{sub}}}{N_{\text{total}}} = \frac{80 \times 500}{2,000} = 20\times \qquad (6)$$

This $20\times$ redundancy minimizes sampling bias and provides stable, reproducible accuracy estimates.

**Testing Conditions.** All participants solved problems individually under strict closed-book conditions. The use of calculators, search engines, or geometry software was strictly prohibited; only pen and paper were permitted for auxiliary sketches. This ensures that the results reflect unaided human reasoning.

**Accuracy Metric.** The reported score is the micro-averaged accuracy over the total response set $\mathcal{R}$ ($|\mathcal{R}| = 40,000$). Let $\hat{y}_i$ be the participant's answer for the $i$-th response and $y_i^*$ be the ground truth. The human accuracy $\text{Acc}_{\text{human}}$ is defined as:

$$\text{Acc}_{\text{human}} = \frac{1}{|\mathcal{R}|} \sum_{i \in \mathcal{R}} \mathbb{I}(\hat{y}_i = y_i^*) = 96.3\% \tag{7}$$

where $\mathbb{I}(\cdot)$ denotes the indicator function, equal to 1 if the condition holds and 0 otherwise.

### B.4 MORE DETAILS ABOUT MATHSPATIAL-SRT

#### B.4.1 PROOFS FOR MATHSPATIAL-SRT

**Proposition 1 (Normal-form coverage).** For every task in MathSpatial-Bench (the 11 subcategories), there exists a finite sequence of operations over $\{\text{CORR}, \text{CONS}, \text{INFER}\}$ that solves the task; moreover, any valid solution can be transformed into an equivalent sequence in the *correlate $\rightarrow$ constrain $\rightarrow$ infer* normal form.

*Proof.*

(i) **Decomposition.** Each subcategory reduces to three stages: - identify correspondences across views (CORR); - enforce projection/geometric rules (CONS); - produce numeric or categorical outputs (INFER). For example, tasks in Holistic Recognition (C3M, 3VM, IVI, CR) require establishing cross-view feature identity and validating composition rules, while Generative Inference tasks (MVC, VT, PRA) require applying projection conventions, and Abstract Deduction tasks (GPC, FPBI, SCP, PPCD) demand property calculation or feasibility inference.

(ii) **Closure.** CORR only augments the correspondence set, while CONS only augments the rule set. Both are monotonic updates, and intermediate queries can be treated as INFER operations that read from the accumulated state without invalidating earlier facts. Thus, any arbitrary solution sequence can be reordered into blocks of correlations, then constraints, followed by inference.

(iii) **Task coverage.** Each of the 11 subcategories can be mapped into this three-stage form, ensuring full expressivity.

**Proposition 2 (Minimality).** The primitive set $\{\text{CORR}, \text{CONS}, \text{INFER}\}$ is minimal: removing any one element strictly reduces expressivity.

*Proof.*

- Without CORR, cross-view identity tasks (e.g., C3M, 3VM, MVC) cannot be solved.

- Without CONS, tasks that require validating geometric or projection rules (e.g., PRA, CR, SCP) cannot be handled.

- Without INFER, tasks requiring numeric calculation or discrete decision outputs (e.g., GPC, FPBI, PPCD) cannot be completed.

Therefore, no primitive is redundant.

We clarify that these proofs apply within the formalized problem space of *MathSpatial*. The claim of minimality is further justified by *semantic orthogonality*: each operation targets a distinct, non-overlapping role—CORR for perceptual alignment, CONS for law application, and INFER for logical deduction. Merging any two would compromise the granularity required for precise error diagnosis (as evidenced in Error Analysis). Furthermore, the practical sufficiency of this set is empirically validated by our OOD generalization results (Section 4.3), which demonstrate that models trained on these primitives successfully transfer to complex, perception-heavy benchmarks (e.g., 3DSRBench-Real).

#### B.4.2 SRT VALIDATION WORKFLOW

To ensure the reliability of GPT-4o–generated SRTs, we employ a dual-agent role-playing validation framework. All traces are grounded by the authoritative official solution, which prevents free-form hallucination and ensures consistency. The Reviewer identifies structural or logical defects, and the Checker rewrites the trace to correct these issues. Failed traces are manually inspected or discarded.

Table 5: Prompt for SRT Reviewer

---

**SRT Reviewer Prompt**

You are a Reviewer for a structured reasoning trace (SRT). Your task is to audit each step for correctness and consistency.

**Rules:**

- Step-by-step verification: check if each step is logically valid.
- Operation-type correctness: CORRELATE / CONSTRAIN / INFER must be used appropriately.
- Consistency: detect contradictions with the authoritative solution.
- No hallucinations: no invented points, edges, or faces.
- Conciseness: flag redundant or unnecessary steps.

**Inputs:** Problem Statement, Diagram Description, Ground-Truth Answer, Official Solution, Candidate SRT.

**Output:** A JSON report:

- `is_valid`: bool
- `issues`: list of {`step_index`, `issue_type`, `explanation`, `suggested_fix`}

---

**Dual-Role Validation Workflow.**    To ensure the reliability of the generated traces, we employ a rigorous validation pipeline anchored by **ground-truth verification**. Crucially, GPT-4o is never tasked with generating solutions from scratch; instead, it reformats verified official solutions into our SRT schema (CORR/CONS/INFER), a constraint that significantly minimizes hallucination. The process operates under a dual-role scheme:

- The **Reviewer (Diagnostic Pass)** first examines every step to produce a structured diagnostic report, checking for logical consistency, operation-type correctness, and identifying any redundancies or hallucinations relative to the official solution.
- The **Checker (Repair Pass)** then synthesizes this report to rewrite the trace. Its primary function is to resolve identified conflicts—fixing operation types, pruning redundant steps, and enforcing the strict CORR → CONS → INFER flow—while ensuring the final derivation matches the ground truth.

Any samples that retain structural conflicts or unstated assumptions after this automated repair are forwarded to human experts for manual rewriting.

**Correction Statistics.**    An analysis of the GPT-4o–generated traces reveals a high initial quality, with approximately **90%** of traces passing the Reviewer's check immediately. The **Auto-Correction** mechanism successfully repairs around **7%** of the data, primarily addressing minor structural issues such as mislabeled operations or formatting errors. The remaining **3%**, which exhibit deeper logical inconsistencies, are routed to trained human annotators for manual correction. Following this multi-stage filtration, a final quality audit on a random sample estimates the residual dataset error rate to be **<1%** (based on a Clopper–Pearson interval with 95% confidence), confirming the robustness of our data curation.

**Prompt Demostration.**    To ensure reproducibility and transparency, we provide the full instructions used to guide our validation agents. Tables 5–6 detail the specific prompt templates for both the Reviewer and Checker roles, including the strict definitions of atomic operations and the error taxonomy used to standardize the diagnostic reports.

## B.5    LIMITATIONS AND FUTURE WORK

Our focus on educational geometry problems, while ensuring low perceptual complexity and high-quality annotations, may limit generalization to real-world scenarios involving natural scenes and complex environments. Additionally, our framework primarily addresses static 2D and basic 3D

Table 6: Prompt for SRT Checker Prompt.

**SRT Checker Prompt**

**Instruction.** You are a Checker and Rewriter. Your task is to repair the SRT based on the diagnostic report while preserving the MathSpatial-SRT schema.

**Task.**

- Fix all issues identified by the Reviewer.

- Ensure strict adherence to CORRELATE / CONSTRAIN / INFER semantics.

- Maintain consistency with the authoritative solution.

- Produce a final answer matching the ground truth.

**Inputs.** Problem statement, diagram, official solution, original SRT, Reviewer diagnostic report.

**Output.**

- `Corrected_SRT`: the repaired reasoning trace,

- `Final_answer`: must match the ground truth,

- `Self_check`: confirmation of consistency.

geometric problems, leaving advanced challenges such as dynamic scene understanding, temporal-spatial relationships, and complex multi-object interactions underexplored.

Future work should prioritize four key directions. First, we aim to extend *MathSpatial* to real-world scenarios, adapting our framework for interactive applications in robotics, VR, and AR while preserving the reasoning-perception separation principle. Second, a critical theoretical objective is to validate the extensibility of our *Structured Reasoning Traces*. While our primitive set ({Correlate, Constrain, Infer}) proves sufficient for idealized geometry, investigating its robustness and potential adaptation against the high noise and unconstrained variables of physical environments is essential. Third, exploring multimodal integration with natural language descriptions and temporal sequences could significantly enhance spatial understanding capabilities. In addition, to safeguard the benchmark's longevity against future data contamination, we also plan to implement the BIG-bench Canary GUID standard, allowing developers to automatically filter our test data from future training corpora. Despite these limitations, *MathSpatial* establishes a robust foundation and systematic framework for advancing spatial reasoning research in multimodal large language models.

## B.6 Social Impact

MathSpatial has the potential to generate significant positive social impact across multiple domains. In education, our framework can enhance AI-powered tutoring systems by providing interpretable spatial reasoning capabilities, helping students better understand geometric concepts through step-by-step explanations. The structured reasoning traces in *MathSpatial-SRT* can serve as educational scaffolding, making complex spatial problems more accessible and supporting personalized learning experiences. Furthermore, our work advances the broader goal of developing more transparent and explainable AI systems, which is crucial for building trust in AI applications across scientific research, engineering design, and educational assessment.

However, several potential negative impacts warrant careful consideration. The automated spatial reasoning capabilities could be misused for academic dishonesty, such as solving standardized test problems or homework assignments without genuine learning. Additionally, if deployed in educational surveillance systems, our technology might raise privacy concerns regarding student data collection and monitoring. To mitigate these risks, we emphasize the importance of responsible deployment, including implementing appropriate safeguards against academic misconduct, ensuring transparent data usage policies, and promoting the use of our framework as a learning aid rather than a replacement for human reasoning development. We encourage future applications to prioritize educational benefit and ethical considerations in their implementation strategies.

### Holistic Recognition

- **Question**：A table has several plates placed on it. From three different angles, the views appear as shown in the diagram. Determine the total number of plates on the table.

top view    front view    left view

A 10    B 11    C 12    D 13

Answer: C
Subcategory: C3M

### Holistic Recognition

- **Question**：As shown in the figure, the main view of a solid composed of several small cubes is?

Answer: C
Subcategory: 3VM

### Holistic Recognition

- **Question**：As shown in the figure, which of the following horizontally placed solids does not have a rectangular left view ?

Answer: B
Subcategory: IVI

### Holistic Recognition

- **Question**：As shown in the figure are the front view and top view of a geometric solid. Then this geometric solid is ?

front view    top view

A. Triangular Prism
B. Cube
C. Triangular Pyramid
D. Rectangular Prism

Answer: A
Subcategory: CR

### Generative Inference

- **Question**：A geometric figure is composed of several identical small cubes. The shapes seen from the front and the top are as shown in the diagram. What is the maximum number of such small cubes needed to construct this geometric figure?

A. 8    B. 9
C.10    D.11

front view   top view

Answer: D
Subcategory: MVC

### Generative Inference

- **Question**：The solid shown in the figure is constructed from 5 identical small cubes. The shape seen from the left side is ?

front view

Answer: D
Subcategory: VT

### Generative Inference

- **Question**：As shown in the figure, the rim of the cup is parallel to the projection plane, and the direction of the projection line is indicated by the arrow. Its orthographic projection is ?

Answer: D
Subcategory: PRA

### Abstract Deduction

- **Question**：The three views of a solid are shown in the figure, where the front view is an equilateral triangle. What is the surface area of the circumscribed sphere of the solid ?

front view    left view    top view

A. $(8/3)\pi$   B. $(16/3)\pi$   C. $(48/3)\pi$   D. $(64/3)\pi$

Answer: D
Subcategory: GPC

### Abstract Deduction

- **Question**：The top view of a geometric figure constructed from five identical small cubes is shown in the diagram. The number of possible ways to arrange this geometric figure is ?

A. 1    B. 2
C. 3    D. 4

Answer: D
Subcategory: FPBI

### Abstract Deduction

- **Question**：As shown in the figure, triangle $\triangle ABC$ is translated along the direction of BC to the position of $\triangle DEF$. If EC = 2BE = 2, then the length of CF is ?

A. 1    B. 2    C. 3    D. 4

Answer: D
Subcategory: SCP

### Abstract Deduction

- **Question**：As shown in the figure, a rectangle with a length of 2 and width of 1 moves horizontally from the left side of line L to the right side in the "posture" shown (the dashed line in the figure is the horizontal line). The distance of the translation is ?

A. 1    B. 2
C. 3    D. $2\sqrt{2}$

Answer: C
Subcategory: PPCD

Figure 9: Example problems from each of the 11 subcategories in **MathSpatial-Bench**, covering the three reasoning levels: *Holistic Recognition*, *Generative Inference*, and *Abstract Deduction*.

