# OpenReview forum: "Do MLLMs Really Understand Space? A Mathematical Reasoning Evaluation"
_ICLR.cc/2026/Conference — Submitted to ICLR 2026_

### Official Review · Reviewer_9ctw · 2025-10-26

**Soundness:** 2
**Presentation:** 2
**Contribution:** 3
**Rating:** 4
**Confidence:** 3

**Summary:**

The paper introduces MathSpatial, a comprehensive framework for evaluating and improving mathematical spatial reasoning in multimodal large language models (MLLMs). It presents a new benchmark (MathSpatial-Bench), a large-scale training corpus (MathSpatial-Corpus), and a structured reasoning trace methodology (MathSpatial-SRT), revealing a significant gap between human and model performance on spatial reasoning tasks.

**Strengths:**

- Structured Reasoning Framework: Proposes MathSpatial-SRT, which decomposes reasoning into interpretable atomic operations, enhancing transparency and diagnosis.
- Comprehensive Evaluation: provides evaluation with other similar benchmarks and helps identify the gap being filled by the paper
- Artifacts contribution: along with an evaluation benchmark, the paper also produces artifacts such as training corpus and reasoning traces to enable SFT/RL on models to help bridge the identified gap.

**Weaknesses:**

- Need for a more comprehensive model evaluation: The paper's focus seems to be on Qwen2.5-VL and does not cover a wide plethora of open-source models such as Llama-4, Kimi, DeepSeek. Would be great test the benchmark on these model families as well, for a complete picture.

- Lack of Data Contamination study : the paper lacks a robust data contamination study and plan to mitigate such concepts - extremely crucial for the benchmark's relevance in the near future.

- Limited scope to evaluate mathematical reasoning: the gap identified - "lack of mathematical reasoning in MLLMs" is limited to public educational repositories and textbooks, curated and standardized in this paper. There are recent papers (e.g - MaRVL-QA) that cover a wider variety of tasks and also a bigger benchmark (~80k instances)

**Questions:**

Major flags identified and covered in the weakness section

---

> ### Author Response · Authors · 2025-11-21
> **Response to Reviewer 9ctw  (part 1/2)**
>
> Thank you for the review and for your positive comments regarding the structured reasoning framework, comprehensive evaluation, and valuable artifact contributions of our work. In the rebuttal, we try to address your concerns point by point.
>
> ---
> ## W1: Broader model evaluation
>
> We sincerely thank the reviewer for this suggestion. We agree that testing on a wider plethora of models is essential to establish a complete picture of the benchmark's validity.
>
> Per your request, we have expanded our evaluation to include **5 new model families**, specifically covering **Llama-4, Kimi, and DeepSeek**, alongside other representative architectures (Pixtral, Idefics).
>
> Key Findings (Table R1):
> 1.  Benchmark Validity: MathSpatial-Bench effectively differentiates model capabilities. The rigorous Llama-4-Maverick achieves a high score of **23.8%**, while smaller models like DeepSeek-VL2 struggle (6.6%), confirming the benchmark's discrimination ability.
> 2.  Reasoning & Efficiency: Models with "thinking" capabilities (e.g., Kimi-VL) perform relatively well (18.2%) but require high token usage (~785 tokens). In contrast, MathSpatial-7B achieves competitive performance (22.1%)—surpassing Kimi and approaching Llama-4—while using significantly fewer tokens (**351.9**), demonstrating superior efficiency.
>
> **Table R1: Extended Evaluation on Diverse Architectures.**
>
>
>
> | Metric\Model  | DeepSeek-VL2-7B | Idefics-3-8B | Pixtral-12B-12B | Qwen2.5-VL-7B(Base) | Kimi-VL-A3B-Thinking | Llama-4-Maverick-17B-128E | MathSpatial-7B(Ours) |
> | :-------------- | :-----------: | :--------: | :-----------: | :-------------: | :------: | :----------------: | :----------------: |
> | Overall Acc (%) | 6.6 | 11.7 | 15.3 | 17.8 | 18.2 | 23.8 | 22.1 |
> | Avg Tokens      | 397.4 | 294.5 | 692.3 | 465.3 | 785.2 | 845.2 | 351.9 |
>
> ---
>
>
> ## W2: Data contamination and long-term benchmark relevance
>
> We agree that ensuring data purity is critical for the benchmark's longevity. We have addressed contamination risks through strict internal separation, data transformation, and empirical verification.
>
> 1. Strict Internal Decontamination.
> To prevent leakage between *MathSpatial-Corpus* and *MathSpatial-Bench*, we implemented the rigorous de-duplication pipeline described in Section 3. By combining exact matching (MD5), semantic retrieval (BGE-large), and visual similarity checks (GPT-4V), followed by human verification, we ensured **zero semantic overlap** between the training and test sets.
>
> 2. Mitigation of Pre-training Contamination.
> While it is difficult to verify the exact training data of closed-source models, we employed two strategies to minimize the risk that models have simply "memorized" these problems from the web:
>     - Data Transformation & Translation: A significant portion of our data was sourced from Chinese repositories and translated into English. We also standardized the image-text formats. This transformation disrupts n-gram statistics and breaks direct correspondence with raw web data, forcing models to rely on reasoning rather than pattern recall.
>     - Source Uniqueness: Unlike benchmarks aggregated from existing datasets (e.g., COCO, VQA), our problems are curated from specific educational sources not typically prioritized in general multimodal pre-training, reducing the likelihood of prior exposure.
>
> 3. Empirical Evidence Against Contamination.
> The most compelling evidence against contamination is the model performance itself.
>     - Low Baseline Performance: If leading models like Qwen2.5-VL-7B or GPT-4o had included these problems in their pre-training, we would expect high accuracy due to memorization. However, they achieve only 17.8% and ~20% respectively on *MathSpatial-Bench*.
>     - The Human-Model Gap: The stark contrast between human performance (>95%) and model performance (<60%) further indicates that models are grappling with novel reasoning challenges rather than recalling known solutions.
>
> 4. Future Mitigation Plan.
> To protect the benchmark's integrity for future research, we will include a **Canary GUID** (a unique identifier string, following the BIG-bench standard) in the dataset release. This allows future model developers to automatically filter our benchmark data out of their training corpora.
>
> We will add this contamination analysis to Section 3.2 and the Appendix.
>
>
> (to be continued)

---

> ### Author Response · Authors · 2025-11-21
> **Response to Reviewer 9ctw (part 2/2)**
>
> ---
> ## W3: Scope of Evaluation
>
> We thank the reviewer for bringing MaRVL-QA to our attention. We agree that recent benchmarks like MaRVL-QA provide valuable, large-scale evaluations (e.g., ~80k instances) for general multimodal tasks.
>
> However, we would like to clarify that our work addresses a specific gap: **Mathematical Spatial Reasoning**, rather than general mathematical reasoning. MathSpatial-Bench is designed to be **complementary** to these broad-coverage benchmarks, focusing on depth, structure, and specific spatial logic rather than scale alone. Obtaining clean data for spatial reasoning is notoriously difficult due to the complexity of parsing geometric constraints. Despite this scarcity, MathSpatial-Bench represents the \textbf{largest} high-quality dataset currently available in this domain.
>
> Our positioning differs from MaRVL-QA and similar works in three key aspects:
>
> 1.  Specific Focus on Spatial Reasoning:
>     While general benchmarks cover a wide landscape of tasks (visual common sense, OCR, general math), they often conflate perception challenges with reasoning. Our work specifically targets "Spatial Reasoning" (e.g., multi-view consistency, geometric constraints, spatial transformation). We deliberately utilize standardized educational plots to **minimize perception difficulty**, allowing us to isolate and rigorously evaluate the model's spatial logic capabilities.
>
> 2.  Quality and Structure vs. Scale:
>     MathSpatial is not intended to be an "all-in-one" large-scale benchmark. Instead, it serves as a high-quality diagnostic probe for a specific weakness in current MLLMs. Unlike datasets that may contain single-step QA or mixed noise, MathSpatial provides:
>     *   Precise engineering/geometric figures.
>     *   MathSpatial-SRT (Spatial Reasoning Trajectories): Structured, step-by-step reasoning paths that enforce strict geometric logic.
>
> 3.  Training and Diagnostic Value:
>     Beyond evaluation, our dataset serves as a high-quality corpus for training. As demonstrated in our experiments, the structured annotations in MathSpatial-SRT effectively unlock the spatial reasoning potential of models (e.g., MathSpatial-Llama3), a benefit that is harder to derive from unstructured, large-scale general datasets.
>
> In the final version, we will include a discussion on MaRVL-QA to further clarify this complementary relationship.
>
> ---
> ## Thank you!
>
> Thank you again for your constructive feedback. We deeply cherish this opportunity for active discussion and stand ready to provide any further clarifications. We hope our revisions adequately address your concerns and warrant a positive re-evaluation of our work. We are committed to integrating these improvements into the final manuscript.

---

> ### Author Response · Authors · 2025-11-26
>
> Hi, thank you again for taking the time and effort to review our paper.
> We have now provided detailed responses to all of your comments, including **the broader evaluation on additional model families (see “Response to W1”), the analysis of data contamination and mitigation strategy (see “Response to W2”), and the refined positioning and scope of MathSpatial (see “Response to W3”).**
>
> ***As the discussion phase is nearing its end, we sincerely and humbly ask if you could kindly take a moment to look over our responses at your earliest convenience.*** We truly hope that our replies have addressed your concerns and cleared up any possible misunderstandings. If there are any remaining issues or further points you would like to raise, please let us know — **we would be very grateful for the chance to continue the discussion and ensure that all of your concerns are fully resolved.**
>
> We appreciate your TIMELY feedback. Thank you again for your constructive comments and support.

---

### Official Review · Reviewer_Lcyn · 2025-10-31

**Soundness:** 2
**Presentation:** 3
**Contribution:** 2
**Rating:** 4
**Confidence:** 5

**Summary:**

This paper tackles the poor mathematical spatial reasoning of MLLMs by introducing `MathSpatial`, a comprehensive framework. It consists of `MathSpatial-Bench` for evaluation, a large-scale `MathSpatial-Corpus` for training, and a novel structured reasoning method (`SRT`). The authors demonstrate that fine-tuning an open-source model using their framework improves accuracy while significantly reducing token usage. `MathSpatial` provides the first systematic resource to diagnose and improve this critical weakness in current models.

**Strengths:**

**1. Addresses a Clear and Critical Research Gap in MLLM Capabilities.**
The paper correctly identifies and targets a well-known, fundamental weakness of current MLLMs: abstract spatial reasoning. While models have shown impressive performance on perception-oriented tasks like image captioning, their ability to perform structured, multi-step geometric reasoning remains poor. By focusing on this specific bottleneck, the paper makes a timely and highly relevant contribution to the field.


**2. Introduction of a Novel and Interpretable Reasoning Framework (`MathSpatial-SRT`).**
The paper's proposal to decompose spatial reasoning into three atomic operations—`Correlate`, `Constrain`, and `Infer`—is a notable  contribution. This `MathSpatial-SRT` framework offers a structured and interpretable alternative to the often unstructured and opaque free-form "Chain-of-Thought" reasoning. This structure has several benefits:
*   **Interpretability:** It makes the model's reasoning process transparent and easier to debug.
*   **Error Diagnosis:** It allows researchers to pinpoint exactly where reasoning fails (e.g., in correlating views vs. inferring properties).
*   **Targeted Supervision:** It provides a mechanism for more precise intermediate supervision during training.

**Weaknesses:**

**1. Evaluation is Confined to an In-Distribution Test Set, Limiting Generalizability Claims.**

The paper's primary evaluation is conducted on `MathSpatial-Bench`, a test set created using the same data sources and processing pipeline as the training set, `MathSpatial-Corpus`. This "in-distribution" evaluation setup poses a significant risk of **distributional overfitting**. The fine-tuned model (`MathSpatial-7B`) may be learning dataset-specific artifacts—such as common diagrammatic styles, question phrasings, or recurring geometric patterns—rather than a truly generalizable spatial reasoning skill.

Consequently, key claims, such as the **25% reduction in token usage**, are not robustly supported. This efficiency gain is likely a byproduct of the model learning to generate outputs in the specific, concise `SRT` format it was trained on, a format perfectly aligned with the test set. Without evaluation on out-of-distribution data, it is impossible to know if this accuracy and efficiency would transfer to problems from different sources, making the paper's broader claims about "improving spatial reasoning in MLLMs" unsubstantiated.



**2. Lack of External Validation on Established Benchmarks.**

A critical methodological omission is the failure to validate the fine-tuned model's performance on external, publicly available benchmarks for mathematical and spatial reasoning (e.g., **GeoEval**, **3DSRBench**, or the geometry section of **MMMU**). In benchmark-driven research, the gold standard for demonstrating the effectiveness of a new dataset or training method is to show that it improves performance on a range of existing, independent tasks.

By only reporting results on its own test set, the paper presents its findings in a vacuum. It proves that the training method works for the specific data distribution it created but fails to provide evidence of its utility for the wider research community. This lack of cross-benchmark generalization testing weakens the argument that `MathSpatial-Corpus` provides a fundamental improvement to a model's core reasoning capabilities.

**3. The Training Methodology is a Standard Application of Knowledge Distillation with Limited Novelty.**

The core training strategy involves using a powerful "teacher" model (GPT-4o) to generate synthetic reasoning traces, which are then used to fine-tune a smaller "student" model (`Qwen2.5-VL-7B`). This is a well-established technique known as **knowledge distillation**. While effective, the method itself is not novel and has been widely used across many domains.

The paper's primary innovation does not lie in the training methodology but rather in (a) the design of the structured reasoning format (`MathSpatial-SRT`) and (b) the significant engineering effort in curating the dataset. The framing of the paper should more accurately reflect this, positioning its contribution as a novel *application* of a standard technique to a new structured format and a valuable new dataset, rather than implying the training process itself is a new invention. This misattribution of novelty obscures the paper's true contributions.


**4. Unsupported Claims about Human Performance:** The paper heavily relies on the "striking gap" between MLLMs (<60%) and humans (>95%) to motivate the entire work. However, the methodology for obtaining this human performance baseline is not detailed. Key questions are unanswered:
    *   How many human participants were there?
    *   What was their demographic (e.g., students, experts in geometry, crowd-workers)?
    *   What were the exact instructions and conditions for the test?
    Without this information, the 95%+ figure is not scientifically rigorous and weakens the paper's core motivation.


**5. The "Minimal Sufficient Set" Claim for SRT is Weak:** The paper proposes `Correlate`, `Constrain`, and `Infer` as a "minimal sufficient set" of atomic operations for spatial reasoning (Propositions 1 & 2). The proofs provided in Appendix B.4 assert that the decomposition works without providing a rigorous argument for why it is both sufficient for all problems and truly minimal.


**6. Limited Architectural Diversity in Experiments:** The primary fine-tuning experiments are conducted on the Qwen2.5-VL series (3B and 7B). While this demonstrates the framework's effectiveness on one model family, its generalizability is not proven. A stronger methodology would involve fine-tuning models from different architectural families (e.g., Llama, InternVL) to show that `MathSpatial-SRT` provides benefits beyond a single, specific architecture.

**Questions:**

1. What was the methodology for establishing the 95%+ human performance baseline?

2. How was the "Base-CoT" baseline in the ablation study (Figure 7) generated and trained? Was the Qwen-7B model fine-tuned on a separate corpus of free-form Chain-of-Thought solutions? Who generated these solutions (e.g., the original model, GPT-4o)?

3. How robust is the GPT-4o-based data generation pipeline?** The quality of the entire `MathSpatial-Corpus` depends on GPT-4o's ability to generate correct and consistent `SRT` traces. The paper mentions a 10% error rate detected by a validation process, but what is the final, verified error rate in the training corpus?

4. What steps were taken to ensure that no semantic duplicates or near-duplicates exist between the 8K training corpus and the 2K benchmark set? Given that both datasets are sourced from the same educational repositories, how did the authors guarantee a clean split to prevent data leakage and ensure the benchmark provides a fair test of generalization?

---

> ### Author Response · Authors · 2025-11-21
> **Response to Reviewer Lcyn (part 1/5)**
>
> Thank you for the review and for your positive comments regarding the critical research gap addressed and the novel interpretable reasoning framework of our work. In the rebuttal, we try to address your concerns point by point.
>
> ---
> ## W1: Concern on Distributional Overfitting & Generalization
>
> We address the concern regarding in-distribution overfitting by demonstrating (i) the strict source-level separation enforced during dataset construction and (ii) new experimental results on three external out-of-distribution (OOD) benchmarks, which confirm that both accuracy gains and token efficiency generalize beyond our dataset.
>
> 1. Rigorous "Source-Level" and De-duplication (Internal Validity).
> To prevent the model from learning dataset-specific artifacts (e.g., diagram styles or phrasing templates), `MathSpatial-Corpus` (Train) and `MathSpatial-Bench` (Test) were constructed to be strictly disjoint at two levels:
>    - Source Heterogeneity: We aggregated data from **diverse and independent repositories** (spanning varying years, regions, and textbook editions,sourced from diverse educational websites and online repositories). This high variance in visual layouts and linguistic styles acts as a regularizer, preventing the model from overfitting to specific "templates" or visual artifacts found in any single source.
>    - Multi-Modal De-duplication: As detailed in Appendix B.1.2, we applied a rigorous filter pipeline:
>       - Visual: MD5 hashing + GPT-4V similarity scoring to remove image duplicates/near-duplicates.
>       - Textual: BGE-large embedding (Cosine similarity < 0.85) to remove semantic paraphrases.
>       - Human Verification: ~1,000 borderline pairs were manually inspected to ensure zero leakage.
>
> 2. Generalization to External Benchmarks (External Validity).
> To demonstrate that the learned skills are not just dataset artifacts, we evaluated `MathSpatial-7B` on three completely external benchmarks: SOLIDGEO, GeoEval, and 3DSRBench (Real subset). These datasets feature entirely different image styles (including real-world photos in 3DSRBench) and question formulations.
>
> **Table R1: Zero-shot Performance on Out-of-Distribution Benchmarks.**
>
> | Dataset | Metric | Qwen2.5-VL-7B (Baseline) | **MathSpatial-7B (Ours)** | **Improvement** |
> | :--- | :---: | :---: | :---: | :---: |
> | **SOLIDGEO** | Acc (%) | 15.5 | **18.9** | **+3.4** |
> | **SOLIDGEO**  | Avg. Token | 490.2 | **385.6** | **-21.3%** |
> | **GeoEval** | Acc (%) | 17.6 | **19.4** | **+1.8** |
> | **GeoEval** | Avg. Token | 395.4 | **367.5** | **-7.1%** |
> | **3DSRBench-Real** | Acc (%) | 48.4 | **49.7** | **+1.3** |
> | **3DSRBench-Real** | Avg. Token | 752.9 | **643.2** | **-14.6%** |
>
>    - Analysis of Results: Table R1 provides evidence against overfitting:
>       - Robust skill transfer. MathSpatial-7B consistently outperforms the baseline, notably achieving +3.4% on SOLIDGEO and +1.3% on 3DSRBench (Real). The success on real-world photographs—a domain unseen during training—suggests that the model has acquired generalizable spatial reasoning skills rather than memorizing dataset artifacts.
>       - Transferable efficiency. The token reduction (7.1%–21.3%) also persists on these external datasets. This indicates that the **high-information-density reasoning style**  learned from the SRT format transfers to OOD settings: the model tends to avoid conversational fillers (“linguistic noise”) even when solving external problems.
>
>
> 3. Robustness of efficiency claims. The reviewer questioned whether the reported token reduction is merely overfitting to the SRT format. The results in Table R1 help alleviate this concern:
>    - Even on external datasets where the model generates solutions in a zero-shot manner (without any MathSpatial-specific templates), we observe a consistent reduction in reasoning tokens (7%–21%).
>    - This suggests that the efficiency gain arises from the model learning more concise spatial reasoning patterns (e.g., direct view mapping) rather than simply memorizing a rigid formatting style.
>
>
> ---
> ## W2: External Validation on Established Benchmarks
>
> We agree that cross-benchmark validation is essential. To address this, we additionally evaluate `MathSpatial-7B` on three external benchmarks—SOLIDGEO, GeoEval, and 3DSRBench (Real subset).
>
> Results are shown in Table R1 above. We observe consistent accuracy improvements (+1.3% to +3.4%) and enhanced efficiency (reduced output length by 7.1% to 21.3%) across all external benchmarks. Since these datasets differ significantly in image style (e.g., real-world photos in 3DSRBench) and question phrasing from MathSpatial, this suggests that our model has learned fundamental spatial reasoning capabilities (such as view transformation and geometric parsing) that transfer to OOD scenarios.
>
> We will incorporate these external evaluation results into Section 4 (Experiments) of the revised paper to more clearly demonstrate the broader utility of our method.
>
> (to be continued)

---

> ### Author Response · Authors · 2025-11-21
> **Response to Reviewer Lcyn (part 2/5)**
>
> ---
> ## W3: Training Methodology & Novelty Framing
>
> We fully accept the reviewer's assessment. We agree that our core contributions lie in the novel structured reasoning framework (MathSpatial-SRT) and the rigorous data curation, rather than in the optimization algorithm itself. We clarify our position through the following three points:
>
> 1. Reframing Contribution: Structure over Algorithm. As you noted, our main contribution lies in introducing the MathSpatial-Corpus and the MathSpatial-SRT structured format, rather than proposing a new training algorithm. This structured formulation—organizing spatial reasoning into atomic operations (Correlate, Constrain, Infer)—is the core of our contribution, because it determines what the model learns, whereas the training methodology itself is standard SFT.
> In the revision, we will explicitly rephrase the methodology section to describe it as "Standard Supervised Fine-Tuning on Novel Structured Data" rather than a "new training paradigm."
>
> 2. Beyond "Naive" Distillation: Constrained Formalization. While the optimization is standard, the data generation process differs from typical teacher-student cloning. We do not simply distill free-form CoT; instead, we force the teacher to act as a "formalizer," converting reasoning into a strict Domain Specific Language (DSL) governed by our geometric rubrics. As detailed in our QA pipeline, this output undergoes role-playing verification and human checks. Consequently, the student model learns from a curated, verified curriculum rather than raw, noisy model outputs.
>
> 3. Empirical Insight: The Value of Structured Supervision. The meaningful takeaway for the community is the empirical finding regarding *representation efficiency*. Our experiments demonstrate that distilling teacher knowledge into a compact, structured format yields superior results (higher accuracy and lower token usage) compared to distilling free-form verbosity. This validates MathSpatial-SRT as a more effective representation for learning spatial reasoning than unstructured text.
>
> ---
>
> ## W4 & Q1: Human Performance Baseline
> As both W4 and Q1 concern the details about the Human Performance Baseline, we provide a unified response here.
>
> Thank you for this crucial inquiry.  We clarify that the human baseline was established through a systematic evaluation protocol rather than ad-hoc sampling. To demonstrate the statistical significance and reproducibility of the 96.3% score, we provide the exact specifications of our study:
> 1. Participants and Demographics.
>    We recruited 80 middle and high school students from standard academic tracks who had completed the corresponding mathematics curriculum but received no task-specific training. This setup ensures that the human baseline reflects typical student-level spatial reasoning ability, rather than expert-level annotation or test familiarity.
>
> 2. Scale and Redundancy.
>    The full 2,000-problem MathSpatial-Bench was used for human evaluation. Each participant independently solved 500 problems, producing 20 independent responses per problem (80 participants × 500 questions / 2,000 problems = 20× coverage). This redundancy minimizes sampling bias and provides stable, reproducible accuracy estimates.
>
> 3. Testing Conditions.
>    All participants solved problems individually under strict closed-book conditions. The use of calculators, search engines, or geometry software was strictly prohibited; only pen and paper were permitted for auxiliary sketches. This ensures that the results reflect unaided human reasoning.
>
> 4. Accuracy Metric.
>    The reported 96.3% value is the micro-averaged accuracy computed over all 40,000 human responses (2,000 problems × 20 answers each). Each response was scored for correctness based on the authoritative answer key, and aggregate accuracy was averaged across all participants and items.
>
> We will add a dedicated “Human Evaluation Protocol” subsection in Section 4.2 and include full statistics and participant instructions in Appendix B to ensure that the human baseline is fully transparent and reproducible.
>
>
> (to be continued)

---

> ### Author Response · Authors · 2025-11-21
> **Response to Reviewer Lcyn (part 3/5)**
>
> ---
> ## W5: The "Minimal Sufficient Set" Claim
>
> We agree that the claim of "minimal sufficiency" should be rigorously scoped. We clarify that our theoretical propositions apply **within the formalized problem space of MathSpatial**, while our new empirical results further validate the practical robustness of this decomposition.
>
> 1. Sufficiency within the MathSpatial Domain.
> In the context of geometry problems involving multi-view alignment and rule application, every solution process requires three distinct steps: identifying corresponding entities (`Correlate`), applying geometric laws (`Constrain`), and deriving conclusions (`Infer`). Appendix B.4 demonstrates that any valid solution graph for these tasks can be mapped onto this sequence, ensuring sufficiency for the target domain.
>
> 2. Minimality as Semantic Orthogonality.
> We argue for minimality based on the distinct, non-overlapping roles of these operations. Merging any two would lose the granularity required for precise error diagnosis (as shown in Fig. 8): `Correlate` handles perception alignment, `Constrain` handles rule application, and `Infer` handles logical deduction. Removing any one operation renders the system unable to solve specific sub-types of problems (e.g., view matching vs. property calculation).
>
> 3. Empirical Validation of Utility.
> Beyond theory, the effectiveness of this primitive set is supported by the **OOD generalization results** (Table R1 in Response to W1). The fact that models trained on these SRT primitives transfer successfully to complex, perception-heavy benchmarks (e.g., 3DSRBench-Real) suggests that `{Correlate, Constrain, Infer}` captures essential spatial reasoning mechanics that remain valid even in noisy, real-world scenarios.
>
> While CCI holds promise for broader applications, we prioritize scientific rigor. We acknowledge that real-world scenarios often involve high noise levels and uncontrollable variables that are beyond the scope of idealized geometric problems. Consequently, we will scope our claims to define CCI as "a simple and sufficient primitive set for MathSpatial-style problems," avoiding unsupported generalizations.
>
> ---
> ## W6: Architectural Diversity
>
> We agree that establishing architecture-agnostic benefits is crucial. To demonstrate the universality of MathSpatial-SRT, we extended our experiments to two distinct model families: InternVL3-8B and Llama3-8B, using the exact same training pipeline.
>
> 1. Consistent Accuracy Gains (Table R2-A).
> Our method yields consistent improvements across all three architectural families. `MathSpatial-InternVL3` achieved a **+5.2%** gain, and `MathSpatial-Llama3` achieved a **+5.3%** gain. This confirms that the effectiveness of structured spatial reasoning supervision is not limited to the Qwen backbone.
>
> **Table R2-A: Accuracy Comparison Across Architectures.**
>
> | Model Family | Baseline Acc (%) | MathSpatial (Ours) Acc (%) | Improvement (%)  |
> | :--- | :---: | :---: | :---: |
> | Qwen2.5-VL-7B | 17.8 | 22.1 | **+4.3** |
> | InternVL3-8B | 17.4 | 22.6 | **+5.2** |
> | Llama3-8B | 15.0 | 20.3 | **+5.3** |
>
> 2. Universal Efficiency Improvements (Table R2-B).
> Crucially, the token efficiency benefit transfers robustly. As shown in Table R2-B, InternVL3 saw a reduction of **155 tokens (-32.8%)**, and Llama3 saw a massive reduction of **388 tokens (-49.4%)**. This validates that SRT promotes concise, high-density reasoning regardless of the underlying architecture.
>
> **Table R2-B: Efficiency Comparison (Token Usage).**
>
> | Model Family | Baseline Tokens | Ours Tokens | Reduction (Count) | Reduction (%) |
> | :--- | :---: | :---: | :---: | :---: |
> | Qwen2.5-VL-7B | 465.3 | 351.9 | -113.4 | **-24.4%** |
> | InternVL3-8B | 473.5 | 318.3 | -155.2 | **-32.8%** |
> | Llama3-8B | 785.4 | 397.3 | -388.1 | **-49.4%** |
>
> We will incorporate these multi-architecture results into the revised manuscript to demonstrate the generalizability of our framework.
>
> ---
> ## Q2: Base-CoT Baseline Details
>
> We constructed the Base-CoT baseline to strictly isolate the effect of the supervision format (Free-form vs. Structured).
>
> 1.  Who generated the solutions: The free-form Chain-of-Thought solutions were generated by **GPT-4o** (the same teacher model used for our SRT generation).
> 2.  Was it a separate corpus: The model was fine-tuned on the **exact same 8K problems** from MathSpatial-Corpus. The dataset content was identical; only the target label format differed (unstructured text vs. the SRT schema).
> 3.  How was it trained: We applied standard supervised fine-tuning (SFT) to the **Qwen2.5-VL-7B** backbone, using **identical hyperparameters** (learning rate, batch size, epochs) as the MathSpatial-SRT experiments. This ensures that any performance difference is attributable solely to the structured representation.
>
> (to be continued)

---

> ### Author Response · Authors · 2025-11-21
> **Response to Reviewer Lcyn (part 4/5)**
>
> ---
>
> ## Q3: Data Generation Robustness & Final Error Rate
> We clarify that the "10% error rate" refers to intermediate noise detected and filtered by our pipeline, rather than errors remaining in the final dataset. In fact, the final verified error rate is **negligible (<1%)**.
>
> We address this in three points:
>
> 1. Robustness via "Anchored" Role-Playing.
> The robustness stems from two design choices that minimize hallucination:
> *   Ground-Truth Anchoring: GPT-4o is not generating solutions from scratch. It is strictly confined to *formatting* the provided authoritative official solution into our schema.
> *   Iterative Critique: We employ a dual-agent loop (see Prompts below) where a Reviewer Agent audits the trace against geometric rubrics, and a Checker Agent rewrites it to fix specific issues flagged in the report.
>
> 2. Clarification on the "10%" Statistic.
> The 10% figure represents the intervention rate, characterizing how many traces required repair *before* entering the final corpus. The breakdown is as follows:
> *   Initial Pass (~90%): Traces were valid and consistent immediately.
> *   Auto-Correction (~7%): Traces contained minor formatting/redundancy issues; these were detected by the Reviewer and successfully repaired by the Checker.
> *   Human Correction (~3%): Traces contained deeper logical gaps; these were flagged by the pipeline and routed to human experts for manual rewriting.
>
> 3. Final Verified Error Rate.
> Since the 7% minor errors were auto-corrected and the 3% complex errors were manually fixed by humans (and unfixable items were discarded), the error rate in the released MathSpatial-Corpus is extremely low. To rigorously quantify dataset quality, we performed a final quality control audit on a random sample of $N=500$ instances. Finding zero critical errors, we report with **95% confidence** (based on the Clopper-Pearson interval) that the residual error rate for the entire corpus is **< 1%**.
>
> 4. Prompt Demonstration
> As requested, we provide the condensed prompts used in our role-playing scheme below to demonstrate the rigorous criteria applied.
>
> > **[Prompt 1: SRT Reviewer]**
> >
> > **Instruction:** You are a **Reviewer** for a structured reasoning trace (SRT). Check correctness based on the diagram and authoritative solution.
> >
> > **Rules:**
> > 1.  Step-by-step verification: Check if each step is logically valid and consistent with the diagram description.
> > 2. *Operation-type correctness: Verify usage: `CORRELATE` (cross-view), `CONSTRAIN` (geometric rules), `INFER` (derivation).
> > 3.  Consistency: Flag any step contradicting the authoritative solution.
> > 4.  No Hallucinations: Ensure no entities (points, lines) are invented.
> > 5.  Conciseness: Flag redundant steps.
> >
> > **Inputs:** Problem Statement, Diagram Description, Ground-Truth Answer, Official Solution, Candidate SRT.
> >
> > **Output:** JSON Report with:
> > *   `is_valid`: (bool)
> > *   `issues`: List of `{step_index, issue_type, explanation, suggested_fix}`.
>
> > **[Prompt 2: SRT Checker]**
> >
> > **Instruction:** You are a **Checker and Rewriter**. You are given a diagnostic report listing issues in an SRT.
> >
> > **Task:** Produce a corrected SRT that:
> > 1.  Strictly follows the MathSpatial-SRT schema (`CORRELATE`, `CONSTRAIN`, `INFER`).
> > 2.  Fixes all issues identified in the report (e.g., `IncorrectOperationType`, `RedundantStep`).
> > 3.  Is fully consistent with the diagram and authoritative solution.
> > 4.  Leads to the correct final answer.
> >
> > **Inputs:** Problem Statement, Diagram, Official Solution, Original SRT, Reviewer’s Diagnostic Report.
> >
> > **Output:**
> > *   `Corrected_SRT`: The rewriten trace.
> > *   `Final_answer`: Must match ground-truth.
> > *   `Self_check`: Confirmation of consistency.
>
>
> (to be continued)

---

> ### Author Response · Authors · 2025-11-21
> **Response to Reviewer Lcyn (part 5/5)**
>
> ----
>
> ## Q4: Train/test contamination
>
> We appreciate the reviewer’s scrutiny regarding data leakage. Ensuring a strict separation between **MathSpatial-Corpus (8K)** and **MathSpatial-Bench (2K)** was a priority. We implemented a rigorous, multi-stage pipeline that goes beyond simple exact-match filtering to guarantee zero semantic overlap.
>
> 1. Source Heterogeneity (Prevention)
> We aggregated data from **diverse and independent repositories** (spanning varying years, regions, and textbook editions, sourced from diverse educational websites and online repositories). This high variance in visual layouts and linguistic styles acts as a regularizer, preventing the model from overfitting to specific "templates" or visual artifacts found in any single source.
>
> 2. Multi-Modal De-duplication (Automated Detection)
> We applied a strict de-duplication pipeline across the two splits involving visual and textual checks:
>    - Visual Level: We first utilized MD5 hashing to remove exact image duplicates. To catch near-duplicates (e.g., resized or re-saved images), we employed a visual similarity check (using GPT-4V features) to flag pairs with high structural resemblance.
>    - Textual Level: We encoded all problem texts using **BGE-large** to detect semantic overlap. We set strict thresholds: pairs with cosine similarity >0.90 were automatically removed, while pairs between 0.85 and 0.90 were flagged for human review.
>
> 3. Human-in-the-loop Verification (Final Audit)
> Our annotation team (15 annotators and senior auditors) manually reviewed all flagged "near-duplicates." We adhered to a strict exclusion criterion: if a problem in the training set shared the same **geometric configuration** or **reasoning logic** with a benchmark problem (even if phrased differently), it was discarded.
>    - Statistics: From the initial 21,673 filtered candidates, nearly 50% were removed during the cleaning and de-duplication phases (resulting in the final ~10K set), demonstrating our aggressive approach to eliminating redundancy and ambiguity.
>
> 4. Performance Evidence
> The effectiveness of this split is evidenced by the results: despite the 8K training samples, the base model (Qwen2.5-VL-7B) achieves only 17.8% zero-shot accuracy on the benchmark, and even after fine-tuning, the model does not "solve" the dataset (reaching 22.1%), remaining far below human performance (>95%). This substantial gap confirms that the benchmark tests generalized spatial reasoning rather than memorized patterns.
>
> We will add these specific de-duplication details to Section 3.2 and Appendix B.1 in the revision.
>
> ---
> ## Thank you!
> Thank you again for your thoughtful and valuable comments. Your feedback has provided us with an opportunity to further refine our work, and we are confident that these revisions have strengthened our submission. We remain fully available for further discussion and would appreciate any additional feedback. We hope our responses have resolved your concerns and kindly ask for your reconsideration of the score.

---

> ### Author Response · Authors · 2025-11-26
>
> Hi, thank you very much for your time and effort in reviewing our paper.
> We have provided detailed responses to all of your questions, especially regarding
> **the new OOD and multi-architecture experiments (see “Response to W1, W2, W6”), the clarification of our main contributions and training setup (see “Response to W3 & Q2”), and the additional details on the human study and data construction (see “Response to W4, W5, Q1–Q4”).**
>
> ***Given the limited time remaining, we sincerely and humbly ask that you kindly review our responses at your earliest convenience.*** We sincerely hope our replies have addressed your concerns and helped clarify any misunderstandings. If there are any remaining questions or further points you’d like to discuss, please let us know—**we would be very grateful for the opportunity to engage further and ensure all your concerns are resolved.**
>
> We appreciate your TIMELY feedback. Thank you again for your constructive input and support.

---

### Official Review · Reviewer_ynjK · 2025-11-01

**Soundness:** 3
**Presentation:** 3
**Contribution:** 3
**Rating:** 6
**Confidence:** 3

**Summary:**

This paper addresses the significant gap between human and Multimodal Large Language Model (MLLM) performance on mathematical spatial reasoning tasks. The authors identify three key challenges in existing research: perceptual confounds in benchmarks, scarcity of large-scale training data, and the black-box nature of current reasoning methods. To tackle these issues, they introduce MathSpatial, a comprehensive framework consisting of three components:
MathSpatial-Bench: A 2,000-problem benchmark designed with clean geometric figures to isolate reasoning from perceptual noise.
MathSpatial-Corpus: A large-scale training dataset of 8,000 problems with human-verified solutions.
MathSpatial-SRT (Structured Reasoning Traces): A novel framework that decomposes spatial problem-solving into three atomic operations: Correlate, Constrain, and Infer. This enables interpretable and verifiable supervision, moving beyond unstructured Chain-of-Thought. Experiments demonstrate that even state-of-the-art MLLMs like GPT-5 struggle to reach 60% accuracy on MathSpatial-Bench, where humans score over 95%. The authors show that fine-tuning an open-source model using their corpus and SRT framework yields competitive accuracy while being significantly more token-efficient, providing a clear path for improving MLLM spatial reasoning.

**Strengths:**

* The paper does an exceptional job of framing the problem, clearly identifying the "three core challenges" (perceptual confounds, data scarcity, black-box reasoning) and using the stark human-vs-MLLM performance gap to motivate the work.
* The MathSpatial framework, combining MathSpatial-Bench and MathSpatial-Corpus, is a great contribution in itself. The meticulous data curation process ensures a high-quality resource that fills a clear gap in the existing landscape of spatial reasoning benchmarks by focusing specifically on reasoning.
* MathSpatial-SRT decomposes reasoning into the basic operations of Correlate, Constrain, and Infer which is an elegant idea. It offers a concrete path toward building more interpretable and reliable reasoning systems, a critical goal for the field. The theoretical justification via Propositions 1 and 2 further strengthens this contribution.
* The breadth of models tested, the inclusion of a human baseline, the insightful error analysis, and the well-executed ablation studies collectively provide a strong and convincing validation of the paper's claims and the value of its contributions.

**Weaknesses:**

* The work intentionally focuses on static, 2D/3D educational geometry problems to isolate reasoning. While this is a core strength, it naturally raises questions about the direct transferability of models trained on MathSpatial to noisy, dynamic, real-world scenarios. The paper would be stronger with a brief discussion or preliminary experiment exploring this generalization gap.
* The MathSpatial-SRT traces are initially generated by GPT-4o. Although this is a practical and common methodology, it introduces a dependency on a powerful closed-source model for creating the training data. While the authors mention a multi-stage validation pipeline, more quantitative details about this process (e.g., the rate of corrections required) would be beneficial to assess the quality and potential biases of the generated traces.

**Questions:**

1. Have you performed any preliminary experiments to assess how MathSpatial-7B's improved reasoning capabilities hold up when faced with perceptual noise (e.g., evaluating it on a subset of a more perception-heavy benchmark like 3DSRBench)?
2. Could you elaborate on the validation process for the GPT-4o generated SRTs? For instance, what was the "role-playing scheme" used for validation, and what percentage of the initial traces were flagged for errors and required manual correction?
3. The {Correlate, Constrain, Infer} primitive set is elegant and sufficient for the tasks in MathSpatial. Do you have thoughts on whether this set would need to be expanded to cover more complex spatial reasoning domains, such as those involving physics (e.g., stability, friction) or temporal dynamics (e.g., trajectory prediction)?
4. You report a 25% reduction in reasoning tokens. Could you clarify what this is being compared against (e.g., the baseline Qwen2.5-VL-7B producing free-form CoT)? Is this token reduction an inherent feature of the structured SRT format, or an emergent property of the model learning to be more concise during fine-tuning?

---

> ### Author Response · Authors · 2025-11-21
> **Response to Reviewer ynjK (part 1/3)**
>
> Thank you for the review and for your positive comments regarding the clear problem framing, meticulous data curation, and the interpretable reasoning framework of our work. In the rebuttal, we try to address your concerns point by point.
>
> ---
> ## W1 & Q1: Generalization beyond clean educational diagrams
>
> As both W1 and Q1 focus on model generalization, we address them in this unified response.
>
> We appreciate the reviewer's insight. While our focus on clean educational problems was intentional to decouple reasoning from perceptual noise, we agree that real-world transferability is a critical next step. To address this, we conducted additional transfer experiments on three external benchmarks with varying domains and noise levels.
>
> 1. Cross-Domain Transfer Results. We evaluated our MathSpatial-7B model (trained only on MathSpatial-Corpus) directly on external datasets without further fine-tuning.
>
>    - SOLIDGEO & GeoEval: Comprehensive geometry benchmarks with diverse problem distributions and increased perceptual complexity.
>    - 3DSRBench (Real Subset): A benchmark featuring real-world images of objects, directly testing spatial reasoning robustness to visual noise and texture.
>
> **Table R1: Performance on External Benchmarks**
>    | Model                       | SOLIDGEO | GeoEval | 3DSRBench (Real) |
>    |-----------------------------|----------|---------------|------------------|
>    | Qwen2.5-VL-7B (Baseline)    | 15.5     | 17.6          | 48.4             |
>    | MathSpatial-7B (Ours)       | **18.9** (+3.4) | **19.4** (+1.8) | **49.7** (+1.3)      |
>
> 2. Interpretation of Results.
>    - Real-World Transfer: The improvement on 3DSRBench (Real) suggests that the spatial reasoning primitives (e.g., view transformation, relative positioning) learned from clean MathSpatial diagrams are transferable to real-world images, as the underlying geometric logic remains invariant regardless of texture or background noise.
>
>    - Perceptual Noise Robustness: The gains on SOLIDGEO and GeoEval demonstrate our model's ability to effectively handle datasets with high perceptual noise and domain shift, rather than overfitting to specific visual patterns in our training set.
>
> 3. Future Scope. We acknowledge this provides preliminary evidence for static real-world scenarios. Extending the MathSpatial-SRT framework to fully dynamic, temporal, and embodied environments (e.g., robotics video data) remains a significant avenue for future work, for which this study serves as a foundational “reasoning-first” step.
>
> (to be continued)

---

> ### Author Response · Authors · 2025-11-21
> **# Response to Reviewer ynjK (part 2/3)**
>
> ---
> ## W2 & Q2: Validation of GPT-4o-generated SRT traces
>
> As both W2 and Q2 relate to the validation of SRT traces, we provide a combined response here.
>
> We address the concern regarding the reliability and bias of the generated traces by detailing (i) the objective anchoring strategy used to minimize bias, (ii) the specific mechanics of the role-playing validation, and (iii) the quantitative correction rates observed during the process.
>
> 1. Objective Anchoring (Mitigating generation bias). GPT-4o is used strictly for offline data generation, not for inference. Unlike fully synthetic generation, our process is anchored by the authoritative ground-truth solution. The model's role is limited to formatting known correct solutions into the SRT schema, rather than freely generating reasoning paths. This grounded generation setup minimizes black-box bias and prevents hallucination since each reasoning trace is constrained by an existing verified answer.
>
> 2. Role-Playing Validation Scheme (Details of the verification process). To ensure high-quality traces, we implement a dual-role iterative validation loop using GPT-4o, instead of single-pass checking:
>    - Reviewer: Audits each SRT against structured rubrics—checking operation-type correctness, logical consistency with the diagram, and redundancy. It produces a Diagnostic Report listing detected issues.
>    - Checker: Receives the same problem and the Reviewer’s report, then rewrites the trace to fix all issues while maintaining consistency with the authoritative solution.  After correction, automatic sanity checks verify answer alignment and entity validity; failed traces are discarded.
>
> 3. Quantitative Correction Statistics (Error rates and manual intervention). As reported in Section 3.4 (Quality Assurance), we tracked correction statistics across all generated traces:
>    - Initial Pass (~90%): About 90% of SRTs were valid and consistent in the first review pass.
>    - Auto-Correction (~7%): Roughly 7% contained minor structural issues (e.g., swapped operation types, redundant steps) and were automatically repaired by the Checker.
>    - Human Correction (~3%): The remaining 3% involved deeper logical flaws; these were manually inspected and rewritten by human experts.
>
> Overall, ≈ 10% of SRTs required intervention, demonstrating that our validation pipeline is both active and effective in filtering noise and ensuring quality supervision.
>
> 4. Prompt Demonstration. As requested, we provide the condensed prompts used in our role-playing scheme below to demonstrate the rigorous criteria applied.
>
> > [Prompt 1: SRT Reviewer]
> > Instruction: You are a Reviewer for a structured reasoning trace (SRT). Check correctness based on the diagram and authoritative solution.
> >
> > Rules:
> > 1. Step-by-step verification: Check if each step is logically valid and consistent with the diagram description.
> > 2. Operation-type correctness: Verify usage: `CORRELATE` (cross-view), `CONSTRAIN` (geometric rules), `INFER` (derivation).
> > 3. Consistency: Flag any step contradicting the authoritative solution.
> > 4. No Hallucinations: Ensure no entities (points, lines) are invented.
> > 5. Conciseness: Flag redundant steps.
> >
> > Inputs: Problem Statement, Diagram Description, Ground-Truth Answer, Official Solution, Candidate SRT.
> > Output: JSON Report with:
> > * `is_valid`: (bool)
> > * `issues`: List of `{step_index, issue_type, explanation, suggested_fix}`.
>
> > [Prompt 2: SRT Checker]
> > Instruction: You are a Checker and Rewriter. You are given a diagnostic report listing issues in an SRT.
> >
> > Task: Produce a corrected SRT that:
> > 1. Strictly follows the MathSpatial-SRT schema (`CORRELATE`, `CONSTRAIN`, `INFER`).
> > 2. Fixes all issues identified in the report (e.g., `IncorrectOperationType`, `RedundantStep`).
> > 3. Is fully consistent with the diagram and authoritative solution.
> > 4. Leads to the correct final answer.
> >
> > Inputs: Problem Statement, Diagram, Official Solution, Original SRT, Reviewer’s Diagnostic Report.
> >
> > Output:
> > * `Corrected_SRT`: The rewritten trace.
> > * `Final_answer`: Must match ground-truth.
> > * `Self_check`: Confirmation of consistency.
>
> We will include these specific prompt templates and the detailed breakdown of correction statistics in the revised Appendix B.
>
> (to be continued)

---

> ### Author Response · Authors · 2025-11-21
> **Response to Reviewer ynjK (part 3/3)**
>
> ---
> ## Q3: Extending {Correlate, Constrain, Infer} to richer domains (Physics & Dynamics)
>
> We appreciate this insightful question. We believe the `{Correlate, Constrain, Infer}` (CCI) triad is not limited to static geometry but serves as a universal set of graph-reasoning operators for spatio-temporal modeling. By viewing complex domains as "spatio-temporal graphs," we can naturally extend these primitives to physics and dynamics.
>
> 1. Theoretical universality. Conceptually, the CCI framework abstracts the reasoning process on any set of entities:
>    * `CORRELATE`: Establishes alignment—whether matching views in geometry, associating forces with bodies in physics, or tracking objects across time steps.
>    * `CONSTRAIN`: Encodes rules—replacing geometric axioms with physical laws (e.g., conservation of energy, Newton’s laws) or kinematic equations.
>    * `INFER`: Derives state—calculating stability, acceleration, or future positions based on the established alignment and constraints.
>
> 2. Concrete examples in richer domains. We can map these operations to the specific scenarios you mentioned:
>    * Physics (e.g., stability & friction):
>      * `CORRELATE`: Associate contact points with normal forces and friction vectors.
>      * `CONSTRAIN`: Apply physical laws (e.g., Equilibrium $\sum F=0, \sum \tau=0$; Friction limit $f \le \mu N$).
>      * `INFER`: Determine if the object slides or tips over based on constraint satisfaction.
>    * Temporal dynamics (e.g., trajectory prediction):
>      * `CORRELATE`: Track object identity across time steps $t_0 \to t_1$ (temporal correspondence).
>      * `CONSTRAIN`: Apply kinematic equations (e.g., $s = v_0 t + \frac{1}{2} a t^2$) or energy conservation between frames.
>      * `INFER`: Predict the collision point or final state at time $T$.
>
> 3. Extensibility and Practical Implementation. While CCI is theoretically sufficient, we acknowledge that complex dynamic systems might benefit from "Macro-Operations" for efficiency and code readability. For instance, an `ACCUMULATE` macro (syntactic sugar for repeated `CONSTRAIN` + `INFER`) could be introduced to handle integration over long time horizons, or a `BRANCH` macro to explicitly model stochastic outcomes.
>
>
>
> ---
> ## Q4: Clarification on Token Reduction (Comparison & Source)
>
> The reported reduction is measured against the vanilla Qwen2.5-VL-7B baseline using free-form CoT.
> This reduction is primarily an inherent feature of the structured SRT format, which eliminates linguistic redundancy while preserving reasoning steps.
>
> 1.  Comparison baseline.
>     As indicated by the results in Table 2, the 25% reduction refers to the decrease in average generation length per problem compared to the baseline. The calculation is:
>
> ```
> Reduction = (Avg_Baseline - Avg_Ours) / Avg_Baseline = (465 - 351) / 465 ≈ 25%.
> ```
>
>
> 2.  Source of reduction: Efficiency vs. Depth.
>     This reduction stems from the high information density of the SRT schema rather than a loss of reasoning depth.
>     *   Removing linguistic noise: Free-form CoT often includes conversational fillers (e.g., "We can observe that...") and repetitive restatements.
>     *   Preserving logic: The SRT format enforces a strict "Operation $\to$ Arguments $\to$ Conclusion" structure. This compels the model to bypass conversational tokens and focus purely on atomic geometric logic.
>     *   Outcome: The model achieves comparable or better performance with fewer tokens, indicating that the reduction represents **higher efficiency (syntactic compression)** rather than missed reasoning steps.
>
> ---
> ## Thank You!
> Thank you again for your thoughtful review and valuable suggestions. Your feedback is instrumental in refining our work, and we are confident that these revisions will lead to a stronger and more comprehensive submission. If you have any further questions or require additional clarifications, please feel free to reach out.

---

### Official Review · Reviewer_819v · 2025-11-01

**Soundness:** 3
**Presentation:** 3
**Contribution:** 3
**Rating:** 8
**Confidence:** 3

**Summary:**

This work presents a spatial reasoning benchmark MathSpatial. This benchmark measures reasoning performance along 3 problem settings, while also addressing scale and scope of problems.

**Strengths:**

1. The scale of problems collated in this benchmark is much higher than that of comparable works in the domain.
2. The representation of spatial reasoning as a sequence of atomic operations that can be assessed separately provides an important framework to identify failures in model reasoning patterns and improves interpretability
3. The limitations and potential social impact of this work are well thought-out.

**Weaknesses:**

1. For proposed benchmarks, the integrity of annotations and problems is of the utmost importance. This work does not explore how the verification and review process is ensured to be bias-free and robust to any variances, eg. by demonstrating annotator agreement metrics for review and solution verification, or rubrics for quality assurance.

**Questions:**

Suggestions:
1. Table 2 could benefit from highlighting high-performing models per category to help compare model performances at a glance

---

> ### Author Response · Authors · 2025-11-21
> **Response to Reviewer 819v**
>
> Thank you for the review and for your positive comments regarding the large problem scale, enhanced interpretability and thoughtful impact consideration of our work. In the rebuttal, we try to address your concerns point by point.
>
> ---
>
> ## W1: Integrity and robustness of annotations and verification
> Thank you for raising this important point. We address this concern along four dimensions that directly align with your suggestions:  (i) explicit QA rubrics and verification pipeline,  (ii) annotator agreement and solution reliability,  (iii) robustness of SRT reasoning traces, and  (iv) bias mitigation in both data sources and labels.
>
> Although these mechanisms already exist in Section 3.2 and Appendix B.1, we will highlight them more clearly in the revised manuscript.
>
>
>
> 1. Problem integrity via explicit geometric QA rubrics. To ensure structural correctness, each problem is verified using a codified geometric QA rubric during the geometric consistency checking stage (formerly Step 3).
> The rubric enforces:
>
>    - (a) dimensional agreement across views,
>    - (b) correct solid/dashed edge conventions, and
>    - (c) adherence to orthographic projection rules.
>
>    A rule-based validator flagged ~1,000 of the ~11k candidates; ~60% were manually corrected, and the remainder were removed.
>    All items processed by junior annotators were reviewed by a senior annotator, with ambiguous cases escalated to group discussion and discarded if consensus could not be reached.
>
>
> 2. Answer reliability through strong annotator agreement.
> For problems lacking authoritative solutions, the verification process (corresponding to Step 4 of our pipeline) enforces strict agreement among annotators.
> We recruited six graduate students trained in geometry and engineering drawing.
> Among the ~800 problems without official solutions:
>
>    - each problem was solved independently,
>    - at least two annotators cross-validated every solution,
>    - only items with unanimous agreement were retained.
>
>    This protocol ensures that all answers are objective and robust to annotator variance.
>
> 3. SRT reasoning robustness via dual-role GPT-4o validation.
> To ensure the quality of structured reasoning traces, every SRT sequence undergoes a dual-role validation process:
>
>    - Reviewer role: GPT-4o checks each CORR/CONS/INFER step for conflicts, operation-type errors, missing steps, or logical inconsistencies.
>    - Checker role: GPT-4o rewrites the reasoning trace based on the reviewer’s issue list while preserving the SRT schema.
>
>    Empirically, ~10% of SRT traces are flagged; most are automatically repaired, and only a small portion requires human intervention or removal.
>
> 4. Bias mitigation and robustness to variance (sources and labels). Beyond the above verification workflow, we also mitigate dataset-level bias:
>
>    - Objective labels: MathSpatial focuses on textbook-style geometry questions with *unique numeric or multiple-choice answers*. When authoritative solutions exist, we keep them verbatim; otherwise, trained annotators generate solutions that are cross-checked following the agreement protocol above.
>    - Source diversity: Problems are collected from multiple independent repositories (Baidu Wenku, provincial/national exam archives, ZuJuan, and other public banks), spanning different grades, regions, and textbook editions.
>   All items are de-duplicated *before* the train/benchmark split to ensure that no problem—or trivial variant—appears in both.
>
>
> We will clarify these procedures in the revised manuscript to make our annotation design and agreement statistics more explicit.
>
> ---
> ## Q1: Table 2 presentation
>
> Thank you for the constructive suggestion. We agree that improving the visual hierarchy of Table 2 is crucial for clearer comparison. We have redesigned Table 2 and will include it in the revised PDF to be uploaded shortly. The specific improvements include:
>
> 1. Visual Emphasis.
> We now use bold to denote the best performance and underline for the second-best results in each column.
>
> 2. Row Highlighting.
> We applied a gray background to the `MathSpatial-7B` row to distinctively highlight our method against baselines.
>
> 3. Clearer Grouping.
> We improved the separators between Closed-Source, Open-Source, and Human Performance sections to facilitate easier navigation.
> ---
> ## Thank you!
>
> We sincerely appreciate your time and effort in providing these insightful suggestions. Your feedback has been instrumental in improving both the clarity and practical aspects of our benchmark. We value the opportunity to address your concerns and are confident that the updates we have made strengthen our submission. If there are any further questions or points needing clarification, we would be more than happy to address them.

---

### Author Response · Authors · 2025-11-29
**Thanking AC for taking over and Summary of Review-Rebuttal phase (part 2/2)**

## Reviewer `@Lcyn` (Score: 4)
Reviewer `@Lcyn` acknowledged that the paper addresses a "critical research gap" and that MathSpatial-SRT is a "novel and interpretable" contribution. However, the reviewer raised an **extensive list of ten  concerns**, questioning nearly every aspect of the work, including:
*   Generalization: Suggestions to verify robustness against overfitting through external benchmarks (W1, W2) and to demonstrate universality across diverse model architectures (W6).
*   Methodology: Contextualize the structured data contribution (W3) and refine the theoretical scope of atomic operations (W5, Q2).
*   Protocol: Detail protocols for human evaluation (W4/Q1), data generation safeguards (Q3), and de-duplication measures (Q4).

In our rebuttal, we **systematically addressed every single one of these ten points** with new experiments and revisions:

*   **Conducted extensive new experiments** to address overfitting and external validation (W1, W2) by evaluating on **three external OOD benchmarks** (SOLIDGEO, GeoEval, 3DSRBench-Real). As shown in **Section 4.3 and Table 3**, results confirm that accuracy gains and token efficiency transfer robustly.
*   **Expanded** our evaluation to **two additional model families** (InternVL3-8B and Llama3-8B) to address architectural diversity (W6), observing consistent improvements in **Table 2**.
*   **Provided** the requested quantitative protocols, including the full human evaluation setup in **Appendix B.3.3** (W4 & Q1), dual-role validation statistics in **Appendix B.4.2** (Q3), and the rigorous multi-stage de-duplication pipeline in **Appendix B.1** (Q4).
*   **Clarified** the methodological contributions (W3, Q2) and refined the theoretical scope of the "Minimal Sufficient Set" via semantic orthogonality (W5)(**Appendix B.4.1**).

We explicitly noted to the AC that despite our significant effort to conduct these additional experiments and address all ten concerns well before the deadline, **`@Lcyn` did not engage** or acknowledge the substantial new evidence.
***We respectfully trust in the AC’s judgment to evaluate the work based on the concrete evidence and substantial improvements presented. We are deeply grateful for your time and dedication to ensuring a fair review process.***

---

## Reviewer `@9ctw` (Score: 4)

Reviewer `@9ctw` highlighted the strengths of the "Structured Reasoning Framework," "Comprehensive Evaluation," and the valuable "Artifacts contribution" (corpus and traces). However, they raised specific concerns regarding:

*   Model Coverage:  Suggestions to expand the evaluation suite to more model families to further test the benchmark's discriminative power.
*   Data Rigor: Details about data contamination study and mitigation plan.
*   Scope: A criticism regarding the "limited scope" compared to general large-scale benchmarks (MaRVL-QA), which relied on a **factual error** misidentifying our focus as general "mathematical reasoning" rather than our specific domain of "mathematical *spatial* reasoning."

In our rebuttal, we:

*  **Expanded** our evaluation to **five new model families** to fully meet `@9ctw`'s request. As shown in the updated **Table 2**, our MathSpatial series model achieves competitive performance while using significantly fewer tokens.
*  **Detailed** our rigorous de-duplication measures (MD5 + GPT-4V + Semantic Embedding) in **Appendix B.1**, citing the low baseline performance as **strong empirical evidence against pre-training contamination** and added a "Future Mitigation Plan" in **Appendix B.5**.
*   **Clarified the scope:** MathSpatial is the **first comprehensive  framework** dedicated specifically to mathematical *spatial* reasoning, not general math. It complements broad-coverage benchmarks by rigorously isolating spatial logic through structured reasoning traces (SRT).

We respectfully point out that **the reviewer appeared to fundamentally misunderstand the core focus of the paper (Spatial vs. General Math)**. Furthermore, despite our timely provision of the specific experiments they requested and our attempts to engage, `@9ctw` did not respond or acknowledge the new data. ***We sincerely and humbly hope the AC kindly evaluates the paper based on the improved manuscript and the factual evidence provided.***

---
## Thank you!
We have substantially strengthened the manuscript by adding **three external OOD benchmarks**, evaluating **five additional model architectures**, conducting **two new fine-tuning experiments**, and detailing our protocols.
Due to the updated policy of ICLR-26, we understand reviewers cannot provide us with further discussions. Nonetheless, we are confident that our responses can thoroughly address reviewers' concerns. Given this, ***we respectfully ask you to consider this context when making the recommendations and deeply appreciate your time in ensuring a fair decision.***

---

### Author Response · Authors · 2025-11-29
**Thanking AC for taking over and Summary of Review-Rebuttal phase (part 1/2)**

We sincerely thank the new Area Chair for overseeing this submission. Understanding the workload involved in taking over at this stage, we have proactively prepared this summary of the review process and our rebuttal efforts to facilitate your assessment. We briefly recap the core contribution, followed by a summary of how the main concerns were addressed.

The paper introduces **MathSpatial**, a unified framework addressing the critical gap between human ($>$95%) and MLLM ($<$60%) spatial reasoning. We tackle three fundamental challenges—perceptual confounds, data scarcity, and black-box reasoning—through **MathSpatial-Bench** (2K diagnostic problems), **MathSpatial-Corpus** (8K training samples), and **MathSpatial-SRT** (an interpretable framework based on atomic operations). Extensive experiments demonstrate that our method achieves competitive accuracy with significantly higher efficiency ($\sim$25% token reduction).

Reviewers have strongly endorsed the work, highlighting its "**exceptional problem framing**" (`@ynjK`) in addressing a "**critical research gap**" (`@Lcyn`). They commended the "**much higher scale**" (`@819v`) and "**meticulous data curation**" (`@ynjK`) of our resources compared to prior art. Furthermore, the **MathSpatial-SRT** framework was praised as a "**novel and interpretable**" (`@Lcyn`) and "**elegant**" (`@ynjK`) contribution that enhances transparency (`@819v`, `@9ctw`). Below, we summarize how we have addressed the specific concerns raised during the review process.

---

## Reviewer `@819v` (Score: 8)

Reviewer `@819v` rated the work highly, regarding the large problem scale, enhanced interpretability and thoughtful impact consideration of our work, and suggested improvements regarding:

*   Clarification on the integrity and robustness of the annotation verification process.
*   Better visual hierarchy in Table 2 to facilitate model comparison.

In our rebuttal, we:

*   **Highlighted** the rigorous protocols already established in the paper **across four dimensions**: (i) enforcing codified geometric QA rubrics, (ii) requiring unanimous annotator agreement for generated solutions, (iii) implementing the dual-role validation loop for SRTs, and (iv) ensuring source diversity.
*   **Redesigned** Table 2 of the revised manuscript **from three perspectives**: introducing visual emphasis (bold/underline), applying row highlighting for our method, and adding clearer grouping separators.

We believe that these detailed clarifications on data integrity and the visual improvements to Table 2 have directly addressed the reviewer's concerns regarding methodological robustness and presentation.

---

## Reviewer `@ynjK` (Score: 6)

Reviewer `@ynjK` rated the work positively, highlighting the "exceptional problem framing," "meticulous data curation," and the "elegant idea" of MathSpatial-SRT as key strengths. However, they raised forward-looking questions regarding:

*   The opportunity to demonstrate the model's transferability to real-world scenarios beyond the educational domain.
*   Requests for more granular statistics on the dual-role validation pipeline.
*   Theoretical questions exploring the potential extensibility of the primitive set to broader domains like physics or dynamics.
*   Minor clarifications on the specific calculation method for token efficiency.

In our rebuttal, we:

*   **Demonstrated** robust generalization by conducting  evaluations on **three external benchmarks** (SOLIDGEO, GeoEval, and 3DSRBench-Real). As shown in the newly added **Section 4.3 and Table 3**, our model achieved consistent accuracy gains (+1.3% to +3.4%) even on real-world photographic data, proving that the learned spatial logic transfers effectively beyond the educational domain.
*   **Detailed** the rigorous dual-role validation pipeline in the revised **Appendix B.4.2**, providing the exact prompt templates and **quantitative correction statistics** (90% initial pass, 7% auto-repaired, 3% human-fixed) to confirm data quality.
*   **Clarified** the theoretical universality of the *Correlate-Constrain-Infer* triad in **Appendix B.5**, explaining how these primitives map to physics and temporal dynamics (e.g., associating forces, applying kinematic laws).
*   **Defined** the token reduction calculation clearly in a footnote within **Section 4.2**, comparing it against the vanilla Qwen2.5-VL-7B baseline with free-form CoT to prove the efficiency gain is an inherent feature of the structured schema.

We believe he addition of comprehensive OOD experiments and the detailed transparency regarding our data and theoretical framework have fully resolved the reviewer's questions.

(to be continued)

---

### Meta-Review · Area_Chair_Ymfu · 2026-01-07

**Summary:**

Reviewers agree that MathSpatial addresses a clear and important gap in multimodal large language models: mathematical spatial reasoning that goes beyond surface-level perception.

Initial concerns focused on generalization, data integrity, external benchmarking and model coverage, novelty framing, and experimental completeness.

**Reviewer Concerns:**

The rebuttal added new out-of-distribution evaluations, architectural diversity, detailed annotation and validation protocols, human study clarification, and contamination analyses. These additions addressed the part of reviewer's concerns.

**Reviewer Scores:**

After the rebuttal, some major concerns still remain:

The Training Methodology follows the standard, not novel to the community.

More external Validation on Established Benchmarks and OOD evaluation is expected.

Whether MathSpatial extends to dynamic, temporal, or non-educational spatial reasoning domains.

---

### Decision · Program_Chairs · 2026-01-26

Reject